# AEGIS: ALMOST SURELY SAFE OFFLINE REINFORCEMENT LEARNING

## ABSTRACT

Ensuring safety guarantees in offline reinforcement learning remains challenging, especially when safety constraints must hold almost surely, i.e., along every possible trajectory. Moreover, as pre-specifying a single safety budget (constraint threshold) is often challenging, it is desirable to learn a foundation policy that can be deployed across a broad range of budgets. We introduce AEGIS (Almost-Sure Epigraph-Guided Implicit Safety), an almost surely safe offline RL framework that can guide diffusion policy training via critics that respect constraints across all feasible budgets. AEGIS characterizes the feasible set of initial state-budget pairs as the epigraph of a feasibility critic updated via the worst-case backup. Building on the proposed characterization, we extend Implicit Q-Learning (IQL) to train both feasibility and reward critics. We use these critics to bias a diffusion policy toward high-value feasible actions. Consequently, AEGIS turns diffusion from a generative prior into a safety-aware controller, enabling a single general policy to respect various budgets without further tuning. Empirical results on the DSRL benchmark and humanoid locomotion tasks show that AEGIS achieves high feasibility with competitive returns, generalizing across feasible constraint thresholds.

## 1 INTRODUCTION

Reinforcement learning (RL) has been successfully applied to various domains such as Atari games, dexterous manipulation, and protein design (Mnih et al., 2015; Andrychowicz et al., 2020; Subramanian et al., 2024). Despite these advancements, deploying RL remains challenging in safety-critical domains such as robotic control, since it does not guarantee safety. To address this issue, a growing body of research has focused on safe RL algorithms (Achiam et al., 2017; Stooke et al., 2020). Moreover, safety constraints in many studies are based on expected costs, which do not guarantee avoidance of catastrophic outcomes under every scenario (Yang et al., 2021; Kim and Oh, 2022). Therefore, ensuring strict adherence to safety conditions *almost surely* across all possible trajectories is indispensable (Sootla et al., 2022; Castellano et al., 2022).

Furthermore, most of the safe RL studies focus on the online setting, which requires online interactions that can be costly or unsafe. This motivates the offline setting, which learns from pre-collected datasets (Levine et al., 2020). Recent work demonstrates that diffusion models naturally fit the offline setting by formulating the policy as a generative model of actions conditioned on states, enabling multimodal behavior (Wang et al., 2023; Hansen-Estruch et al., 2023; Kang et al., 2023; Zheng et al., 2024).

A further challenge arises in defining these safety conditions: determining appropriate constraint thresholds beforehand is often challenging. For instance, consider a robotic system where prolonged issuing high-magnitude torque commands may overheat motors or damage joints. Constraining energy cost can help avoid such failures, but the choice of threshold is critical. If the threshold is set too high, failures may still occur; if it is set too low, achievable return may be reduced. The appropriate threshold depends on several factors, such as the hardware condition, the surrounding temperature, and the type of task. Consequently, setting a single fixed value for all situations would be impractical. This motivates the development of methods that can ensure safety for any feasible threshold specified after training (Lee et al., 2023; Guo et al., 2025).

To address these issues, we propose AEGIS (Almost-sure Epigraph-Guided Implicit Safety), a safe offline RL method designed to satisfy an almost-sure constraint for any feasible thresholds. We analyze the feasibility of almost surely safe RL by defining feasibility critics whose epigraph captures the feasible set. These critics are updated using our proposed feasibility Bellman operator, which captures worst-case scenarios over future transitions to ensure safety. Building on implicit Q-learning (IQL) (Kostrikov et al., 2022), we train these critics offline with the expectile-based approximation that captures the operator's risk sensitivity. Guided by the feasibility and reward critics, the diffusion policy samples actions that are both feasible and high-reward. We demonstrate the safety and performance of our approach on the DSRL benchmark (Liu et al., 2024) and humanoid locomotion.

Our main contributions are as follows:

- We investigate the feasibility of almost surely safe RL, introducing feasibility critics whose epigraph characterizes the feasible set of state-budget pairs.
- We present AEGIS, a diffusion-based offline safe RL framework that samples actions respecting risk-sensitive safety across a wide range of safety budgets.
- AEGIS improves the feasibility rate by an average of 18%p across DSRL tasks and demonstrates adaptive behavior in the humanoid locomotion task.

## 2 PRELIMINARY

### 2.1 ALMOST SURELY SAFE REINFORCEMENT LEARNING

Safe reinforcement learning (RL) (Achiam et al., 2017; Stooke et al., 2020) can be represented within the constrained Markov decision process (CMDP) (Altman, 1998) framework. A CMDP is a tuple $(\mathcal{S}, \mathcal{A}, \mathcal{P}, R, C)$, where $\mathcal{S}$ is a state space, $\mathcal{A}$ is an action space, $\mathcal{P} \colon \mathcal{S} \times \mathcal{A} \to \Delta(\mathcal{S})$ is a transition model, $R \colon \mathcal{S} \times \mathcal{A} \to \mathbb{R}$ is a reward function, and $C \colon \mathcal{S} \times \mathcal{A} \to [0, 1]$ is a cost function. We assume that $R$ is bounded, and for technical convenience, $S$ and $A$ are finite and non-empty. We can define almost surely safe RL (Sootla et al., 2022; Castellano et al., 2022) as:

$$
\max_{\pi} \mathbb{E}_{\pi} \left[ \sum_{t=0}^{\infty} \gamma^t R(s_t, a_t) \right]
$$
$$
\text{s.t.} \sum_{t=0}^{\infty} \gamma^t C(s_t, a_t) \leq \delta_0 \text{ a.s.,}
\tag{1}
$$

where "a.s." denotes "almost surely," ensuring that the cost return does not exceed the threshold $\delta_0$ for any trajectory. Unlike in unconstrained RL, selecting an optimal action based solely on the current state is insufficient, as constraint feasibility depends on the cost incurred so far. Accordingly, the policy $\pi$ must also consider the remaining budget $\delta_t = \frac{\delta_0 - \sum_{i=0}^{t-1} \gamma^i C(s_i, a_i)}{\gamma^t}$.

Since the cost return is bounded as $\sum_{t=0}^{\infty} \gamma^t C(s_t, a_t) \leq \frac{1}{1-\gamma}$, we clip the remaining budget from above at $\frac{1}{1-\gamma}$ without loss of generality. Define the budget $\delta_t = \min\left(\frac{\delta_0 - \sum_{i=0}^{t-1} \gamma^i C(s_i, a_i)}{\gamma^t}, \frac{1}{1-\gamma}\right)$, which follows the transition:

$$
\delta_{t+1} = \min\left(\frac{\delta_t - C(s_t, a_t)}{\gamma}, \frac{1}{1-\gamma}\right).
\tag{2}
$$

Following Sootla et al. (2022), the problem (1) can be reformulated using an augmented MDP $(\tilde{\mathcal{S}}, \mathcal{A}, \tilde{\mathcal{P}}, R)$:

$$
\max_{\pi} \mathbb{E}_{\pi} \left[ \sum_{t=0}^{\infty} \gamma^t R(s_t, a_t) \right]
$$
$$
\text{s.t.} \ \delta_t \geq 0 \text{ a.s.,} \ \forall t \geq 0,
\tag{3}
$$

where the augmented state space $\tilde{\mathcal{S}} = \left\{ (s, \delta) \middle| s \in \mathcal{S}, \delta \in \left[ -\frac{1}{1-\gamma}, \frac{1}{1-\gamma} \right] \right\}$ consists of state-budget pairs. Here, the augmented transition model $\tilde{\mathcal{P}}$ maps $((s_t, \delta_t), a_t)$ to a distribution over $(s_{t+1}, \delta_{t+1})$,

where $s_{t+1} \sim \mathcal{P}(s_t, a_t)$ and $\delta_{t+1}$ updates as (2). From the definition of the budget $\delta_t$, the constraint in (3) is equivalent to (1). We denote this constraint as the *almost-sure constraint*. Throughout the paper, the constraint refers to the almost-sure constraint, unless noted otherwise. Sootla et al. (2022) also converts the constrained problem into an unconstrained one by replacing this constraint with an infinitely large reward penalty. In practice, finite reward penalties are typically employed.

## 2.2 DIFFUSION POLICY

One of the successful approaches to applying diffusion models in offline reinforcement learning is to replace the policy with a diffusion process that samples actions conditioned on the current state (Wang et al., 2023; Hansen-Estruch et al., 2023). This generative framework effectively captures the data distribution and enables flexible, multimodal behavior. Kang et al. (2023); Ding et al. (2024) train the noise model $\mu$ of the diffusion policy to match the weighted data distribution: $\pi(a|s) \propto w(s, a)p_{\mathcal{D}}(a|s)$, where $p_{\mathcal{D}}$ is the behavior policy and $w(s, a)$ is the importance weight that guides the diffusion policy toward sampling actions with higher expected returns.

## 2.3 IMPLICIT Q-LEARNING

Implicit Q-Learning (IQL) (Kostrikov et al., 2022) is an offline reinforcement learning method that avoids explicitly querying the Q-function for out-of-distribution actions. IQL learns a value function through expectile regression on the Q-functions of in-dataset actions, which enables value learning without an explicit policy. It has been established that this implies an implicit policy within the learned value function, which can be effectively extracted using diffusion-based policy extraction methods (Hansen-Estruch et al., 2023).

## 3 FEASIBILITY FOR ALMOST-SURE CONSTRAINT

Since diffusion models generate samples from the distribution that matches the training data, they are capable of sampling actions consistent with the behavior policy. The remaining design goals in our settings are: (1) constraining actions to satisfy constraints, and (2) guiding the policy toward high reward returns. In this section, we analyze the feasibility of the almost-sure constraint to address the former issue.

### 3.1 FEASIBLE SET AND POLICY FEASIBILITY

The feasibility of the almost-sure constraint in (3) depends on the initial state $s_0$ and budget $\delta_0$. That is, a policy that is feasible starting from one pair $(s_0, \delta_0)$, may violate the constraint starting from another. Moreover, there may exist initial augmented states $(s_0, \delta_0)$ from which no policy can satisfy the constraint. We define the *feasible set $\mathcal{F} \subset \tilde{\mathcal{S}}$* as the set of all augmented states from which some policy satisfies the constraint. For any state $s$, $(s, \delta)$ with the budget $\delta = \frac{1}{1-\gamma}$ belongs to the feasible set $\mathcal{F}$ since costs are bounded. We next show that Problem (3) need not be solved separately for each initial state-budget pair (or any distribution over them):

**Proposition 3.1.** *There exists a policy $\pi^*$ that is a common solution to the constrained optimization problem (3) for all initial state-budget pairs in the feasible set $\mathcal{F}$.*

Hence, it suffices to consider policies that satisfy the constraint for all $\tilde{s}_0 \in \mathcal{F}$; we call such policies *feasible*. Since $\mathcal{F}$ can be viewed as the viability kernel (Aubin and Saint-Pierre, 2007), we call an action $a$ *viable* at $\tilde{s}$ if the next augmented state $\tilde{s}' \sim \tilde{\mathcal{P}}(\cdot \mid \tilde{s}, a)$ lies in $\mathcal{F}$ almost surely. This yields the following proposition, which restates the classical result from the viability theory in our augmented MDP:

**Proposition 3.2.** *An augmented state $\tilde{s}$ is in the feasible set $\mathcal{F}$ if and only if there exists an action viable at $\tilde{s}$. A policy $\pi$ is feasible if and only if, for every $\tilde{s} \in \mathcal{F}$, $a \sim \pi(\cdot|\tilde{s})$ is viable at $\tilde{s}$ almost surely.*

This characterization reduces the constraints over an infinite time horizon to a *single-step* condition once the feasible set $\mathcal{F}$ is identified. Proofs of Propositions 3.1 and 3.2 are provided in the Appendix B.1 and B.2.

### 3.2 Feasible Set as Epigraph of Feasibility Value Function

From the definition of the almost-sure constraint, the initial point $(s, \delta_A)$ with a larger threshold $\delta_A > \delta_B$ is also in the feasible set $\mathcal{F}$, if $(s, \delta_B) \in \mathcal{F}$. Thus, the feasible set $\mathcal{F}$ can be characterized by identifying the minimal $\delta_0$ for $s_0 \in \mathcal{S}$ such that $(s_0, \delta_0)$ belongs to the feasible set. We define *feasibility value function* $V_f : \mathcal{S} \to \mathbb{R}_0^+$ as follows:

$$V_f(s) = \inf\{\delta | (s, \delta) \in \mathcal{F}\}. \tag{4}$$

In other words, the feasible set $\mathcal{F}$ is the epigraph of the feasibility value function $V_f$. Note that a lower value of $V_f(s)$ indicates a wider range of feasible budgets, as it reflects the minimal budget required to make state $s$ feasible. Similarly, we define the *feasibility Q-function* $Q_f : \mathcal{S} \times \mathcal{A} \to \mathbb{R}_0^+$, a novel critic that enables single-step evaluation of action feasibility under the almost-sure constraint:

$$Q_f(s, a) = \inf\{\delta | (s', \delta') \in \mathcal{F} \text{ a.s. for } (s', \delta') \sim \tilde{\mathcal{P}}(\cdot | (s, \delta), a)\}, \tag{5}$$

$$= \operatorname*{ess\,sup}_{s' \sim \mathcal{P}(\cdot | s, a)} [C(s, a) + \gamma V_f(s')], \tag{6}$$

where $\operatorname{ess\,sup}$ denotes essential supremum—the least upper bound, ignoring a measure-zero set. The derivation from (5) to (6) is provided in the Appendix B.3. We can now reformulate the almost-sure constraint as:

$$Q_f(s, a) \leq \delta \text{ a.s. for all } (s, \delta) \in \mathcal{F}, \tag{7}$$

where $a \sim \pi(\cdot | s, \delta)$. This implies that a feasible policy should sample actions with $Q_f(s, a) \leq \delta$ almost surely, whenever $(s, \delta) \in \mathcal{F}$.

### 3.3 Feasibility Bellman operator

The feasibility Q-function defined in (6) resembles the standard Q-function, except for the use of *essential supremum* over transition dynamics. This captures the worst-case scenario rather than "lucky" transitions to ensure strict constraints. We define the feasibility Bellman operator $\mathcal{T}_f$ to extend the Bellman equation to the feasibility Q-function:

$$\mathcal{T}_f Q_f(s, a) = \operatorname*{ess\,sup}_{s' \sim \mathcal{P}(\cdot | s, a)} [C(s, a) + \gamma V_f(s')],$$
$$V_f(s') = \min_{a'} Q_f(s', a'). \tag{8}$$

**Theorem 3.3.** *Iterating the feasibility Bellman operator $\mathcal{T}_f$ converges to the unique fixed point, which is the feasibility Q-function.*

A proof of the Theorem 3.3 is provided in the Appendix B.4. This novel operator reflects the worst-case transitions via the essential supremum over one step, which is simple yet crucial for ensuring safety almost surely.

## 4 AEGIS: Almost-Sure Epigraph-Guided Implicit Safety

Leveraging the feasibility Bellman operator (8), this section introduces AEGIS, a practical framework for offline safe RL. We approximate the essential supremum with an expectile and extend Implicit Q-Learning (IQL) (Kostrikov et al., 2022) to learn both a feasibility critic and a constraint-aware reward critic. Guided by these critics, we train a diffusion policy that generalizes across budgets by sampling high-reward feasible actions.

### 4.1 Expectile Approximation of Feasibility Bellman Operator

Directly computing the essential supremum is often impractical. To address this limitation, we propose an approximation of the operator (8) using the expectile:

$$\mathcal{T}_f^\alpha Q_f^\alpha(s, a) = \mathbb{E}^\alpha_{s' \sim \mathcal{P}(\cdot | s, a)} [C(s, a) + \gamma V_f^\alpha(s')],$$
$$V_f^\alpha(s') = \min_{a'} Q_f^\alpha(s', a'). \tag{9}$$

where $\mathbb{E}^\alpha$ denotes the $\alpha$-expectile, expectile at level $\alpha \in (0,1)$, of the distribution. Here, $\alpha$-expectile with $\alpha > 0.5$ emphasizes worst-case scenarios and converges to essential supremum as $\alpha$ approaches 1. Since $\mathcal{T}_f^\alpha$ is also a $\gamma$-contraction, iterating it converges to the unique fixed point $Q_f^\alpha$, which defines an approximation of the feasibility Q-function. As shown in Appendix B.5, the fixed point $Q_f^\alpha$ converges to $Q_f$ as $\alpha$ approaches 1. Appendix C demonstrates the accuracy of this approximation in a simple toy environment.

In contrast to many risk-sensitive safe RL methods that define risk measures over policy-dependent cost return, our formulation $Q_f^\alpha$ applies the expectile only to the dynamics backup, thereby characterizing feasibility in a policy-agnostic view. This separation is particularly beneficial in offline settings, where the behavior policy can be diverse, suboptimal, or even unsafe.

## 4.2 Augmented Reward with Penalty

Recall that feasible (viable) action $a$ should satisfy $Q_f(s,a) \leq \delta$ so that almost surely $V_f(s') \leq \delta'$. Unfortunately, $Q_f^\alpha(s,a) \leq \delta$ with $\alpha < 1$ does not guarantee $V_f(s') \leq \delta'$ or $V_f^\alpha(s') \leq \delta'$. To mitigate this relaxation, we also introduce a reward penalty for $V_f^\alpha(s) > \delta$:

$$
\tilde{R}(s,\delta,a) = \begin{cases} R(s,a), & V_f^\alpha(s) \leq \delta, \\ R(s,a) - n, & V_f^\alpha(s) > \delta, \end{cases} \tag{10}
$$

where $\tilde{R} \colon \tilde{\mathcal{S}} \times \mathcal{A} \to \mathbb{R}$ is augmented reward function and $n$ is a hyperparameter for the scale of the reward penalty. With this double safety strategy, consider the following constrained optimization:

$$
\max_\pi \mathbb{E}_\pi \left[ \sum_{t=0}^\infty \gamma^t \tilde{R}(s_t, \delta_t, a_t) \right]
$$
$$
\text{s.t. } Q_f^\alpha(s,a) \leq \delta \text{ a.s. for all } (s,\delta) \in \mathcal{F}. \tag{11}
$$

**Theorem 4.1** (Informal). *If either $\alpha \to 1$ or $n \to \infty$, the optimal policy $\pi^*$ guaranteed by Proposition 3.1 is also optimal for (11) starting from feasible initial state and budget.*

A precise statement and its proof are provided in the Appendix B.6. While Sootla et al. (2022) also employ reward penalties to enforce almost-sure safety, our method differs by incorporating a feasibility critic $V_f$ that anticipates worst-case future costs, not just the current remaining budget. This forward-looking perspective enables earlier penalization of potentially unsafe trajectories, allowing our approach to achieve high safety with practical penalty $n$.

## 4.3 Offline Training of Feasibility and Reward Critics

We estimate the feasibility critics $V_f^\alpha$ and $Q_f^\alpha$ using neural networks $V_f^\psi$ and $Q_f^\theta$ with parameters $\psi$ and $\theta$, respectively. Here, $\psi$ and $\theta$ denote network parameters and should not be confused with the expectile level $\alpha$, which is omitted for clarity. We train these feasibility critic networks using operator (9), entirely from the fixed dataset $\mathcal{D}$. The feasibility Q-network $Q_f^\theta$ can be trained using the expectile loss:

$$
\mathcal{L}_{Q_f}(\theta) = \mathbb{E}_{(s,a,s')\sim\mathcal{D}} \left[ w_{Q_f} u_{Q_f}^2 \right], \tag{12}
$$
$$
u_{Q_f} = C(s,a) + \gamma V_f^\psi(s') - Q_f^\theta(s,a), \tag{13}
$$
$$
w_{Q_f} = |\alpha - \mathbb{1}\{u_{Q_f} < 0\}|, \tag{14}
$$

which assigns an asymmetric weight $w_{Q_f}$ to emphasize the worst-case tail. Inspired by implicit Q-learning (IQL) (Kostrikov et al., 2022), the feasibility value network $V_f^\psi$ is trained with loss:

$$
\mathcal{L}_{V_f}(\psi) = \mathbb{E}_{(s,a)\sim\mathcal{D}} \left[ w_{V_f} u_{V_f}^2 \right], \tag{15}
$$
$$
u_{V_f} = Q_f^\theta(s,a) - V_f^\psi(s), \tag{16}
$$
$$
w_{V_f} = |\tau_f - \mathbb{1}\{u_{V_f} < 0\}|, \tag{17}
$$

---

**Algorithm 1** AEGIS: Almost-Sure Epigraph-Guided Implicit Safety

---

**Input:** Dataset $\mathcal{D} = \{(s,a,r,c,s')\}$; discount factor $\gamma$; expectile levels $\alpha$, $\tau_f$, and $\tau_r$; reward penalty $n$; inverse temperature $\beta$; learning rate $\eta$
**Output:** Safe diffusion policy $\mu_\phi$ and critics $Q_f^\theta, V_f^\psi, Q_r^\zeta, V_r^\xi$

1: **for** each iteration **do**
2:     Sample $(s,a,r,c,s') \sim \mathcal{D}$
3:     $\theta \leftarrow \theta - \eta\nabla_\theta \mathcal{L}_{Q_f}$                                                                                ▷ (12)
4:     $\psi \leftarrow \psi - \eta\nabla_\psi \mathcal{L}_{V_f}$                                                                                ▷ (15)
5:     Sample budget $\delta \sim \mathcal{U}(0, \frac{1}{1-\gamma})$
6:     $\zeta \leftarrow \zeta - \eta\nabla_\zeta \mathcal{L}_{Q_r}$                                                                                ▷ (18)
7:     $\xi \leftarrow \xi - \eta\nabla_\xi \mathcal{L}_{V_r}$                                                                                ▷ (20)
8:     Sample timestep $k$ and noise $\epsilon \sim \mathcal{N}(0, I)$
9:     $\phi \leftarrow \phi - \eta\nabla_\phi \mathcal{L}_\mu$                                                                                ▷ (23)
10: **end for**

---

so that $V_f^\psi$ tracks the feasibility Q-functions from safe actions. Here, $\tau_f \in (0,1)$ is the expectile level for implicit feasibility value learning, where $\tau_f < 0.5$ emphasizes safe actions.

We similarly approximate the reward critics, which refer to the maximum constraint-satisfying expected reward return, with networks $V_r^\xi$ and $Q_r^\zeta$, parameterized by $\xi$ and $\zeta$, respectively. Note that the reward value function should not account for the reward Q-function of infeasible actions. Thus, the reward critics must consider feasibility and, consequently, depend on the current budget. We train them using the loss:

$$\mathcal{L}_{Q_r}(\zeta) = \mathbb{E}_{(s,a,s')\sim\mathcal{D}, \delta\sim\mathcal{U}(0,\frac{1}{1-\gamma})}\left[u_{Q_r}^2\right], \tag{18}$$

$$u_{Q_r} = \tilde{R}_n(s,\delta,a) + \gamma V_r^\xi(s',\delta') - Q_r^\zeta(s,\delta,a), \tag{19}$$

$$\mathcal{L}_{V_r}(\xi) = \mathbb{E}_{(s,a)\sim\mathcal{D}, \delta\sim\mathcal{U}(0,\frac{1}{1-\gamma})}\left[w_{V_r}u_{V_r}^2\right], \tag{20}$$

$$u_{V_r} = Q_r^\zeta(s,\delta,a) - V_r^\xi(s,\delta), \tag{21}$$

$$w_{V_r} = \begin{cases} |\tau_r - \mathbb{1}\{u_{V_r} < 0\}|, & Q_f^\theta(s,a) \leq \max(\delta, V_f^\psi(s)), \\ |\tau_r - 1|\mathbb{1}\{u_{V_r} < 0\}, & Q_f^\theta(s,a) > \max(\delta, V_f^\psi(s)), \end{cases} \tag{22}$$

considering feasibility Q-function. For feasible actions, the loss follows expectile weighting as in IQL. For infeasible actions, we selectively remove their contribution by adjusting the weights to zero. However, to preserve the monotonicity of the optimal reward value with respect to $\delta$, we retain the impact of low-reward actions, even when they are infeasible.

### 4.4 DIFFUSION POLICY

We train a state-conditioned diffusion model, denoted by $\mu_\phi$ with parameter $\phi$, that predicts noise at each denoising step given the current augmented state $\tilde{s} = (s, \delta)$ and diffusion timestep $k$. We aim to sample actions satisfying the constraint $Q_f^\theta(s,a) \leq \max(\delta, V_f^\psi(s))$ while exhibiting high reward advantage $Q_r^\zeta(s,\delta,a) - V_r^\xi(s,\delta)$. To achieve this, we guide the diffusion training using a weight $w(s,\delta,a)$:

$$\mathcal{L}_\mu(\phi) = \mathbb{E}_{k\sim\mathcal{U}(1,T), \epsilon\sim\mathcal{N}(0,I), (\tilde{s},a)\sim\tilde{\mathcal{D}}}\left[w(\tilde{s},a)\left\|\epsilon - \mu_\phi\left(\sqrt{\bar{\alpha}_k}a + \sqrt{1-\bar{\alpha}_k}\epsilon, \tilde{s}, k\right)\right\|\right], \tag{23}$$

$$w(s,\delta,a) = \begin{cases} \exp(\beta(Q_r^\zeta(s,\delta,a) - V_r^\xi(s,\delta))), & Q_f^\theta(s,a) \leq \max(\delta, V_f^\psi(s)), \\ 0, & Q_f^\theta(s,a) > \max(\delta, V_f^\psi(s)), \end{cases} \tag{24}$$

where $(s,\delta,a) \sim \tilde{\mathcal{D}}$ denotes hindsight sampling $(s,a) \sim \mathcal{D}$ and $\delta \sim \mathcal{U}(0, \frac{1}{1-\gamma})$. Here, $\bar{\alpha}_k$ is the cumulative noise-schedule product from the diffusion process and $\beta$ is the inverse temperature. This formulation enforces feasibility by assigning zero weight to constraint-violating actions, while softly prioritizing high-advantage actions through an exponential scaling. As a result, the diffusion model learns to denoise toward actions that are both safe and effective, without requiring any explicit projection. The complete algorithm for training AEGIS is described in Algorithm 1. During inference, we generate the action using the trained diffusion policy $\mu_\phi$. Additional details appear in Appendix D.

Table 1: Results for the feasibility rate (FR) and constrained reward return (CRR) on the DSRL benchmark. Higher is better. Each value is averaged over 50 evaluation episodes and three distinct random seeds. **Bold**: safe agents with FR $\geq 0.75$. Gray: unsafe agents. **Blue**: best safe agent(s) for each task.

| TASK | BCQ-Lag | | COptiDICE | | $\gamma$-CDT | | TREBI | | IDQL-Sauté | | Ours | |
|---|---|---|---|---|---|---|---|---|---|---|---|---|
| | FR ↑ | CRR ↑ | FR ↑ | CRR ↑ | FR ↑ | CRR ↑ | FR ↑ | CRR ↑ | FR ↑ | CRR ↑ | FR ↑ | CRR ↑ |
| CarButton1 | 0.41 | 0.24 | 0.58 | 0.15 | 0.42 | 0.24 | 0.47 | 0.26 | 0.45 | 0.15 | **0.79** | **0.10** |
| CarGoal1 | 0.69 | 0.48 | 0.65 | 0.49 | **0.87** | **0.39** | 0.57 | 0.52 | 0.45 | 0.46 | **0.95** | **0.18** |
| PointButton1 | 0.43 | 0.30 | 0.51 | 0.23 | 0.53 | 0.24 | 0.53 | 0.25 | 0.50 | 0.17 | **0.80** | **0.12** |
| PointGoal1 | 0.56 | 0.59 | 0.58 | 0.41 | **0.81** | **0.36** | 0.67 | 0.53 | 0.40 | 0.50 | **0.75** | **0.18** |
| HalfCheetah Vel | 0.07 | 0.87 | **1.00** | **0.59** | 0.21 | 0.40 | **1.00** | 0.35 | **1.00** | 0.33 | **1.00** | **0.84** |
| Swimmer Vel | 0.01 | 0.09 | 0.15 | 0.27 | **0.99** | **0.41** | 0.00 | - | **0.95** | 0.05 | **0.91** | 0.20 |
| SafetyGym Avg. | 0.36 | 0.43 | 0.58 | 0.36 | 0.64 | 0.34 | 0.54 | 0.32 | 0.63 | 0.28 | **0.87** | **0.27** |
| AntRun | 0.07 | 0.36 | **0.92** | **0.61** | 0.16 | 0.72 | 0.30 | 0.70 | **0.99** | 0.68 | **1.00** | **0.71** |
| AntCircle | 0.35 | 0.49 | 0.50 | 0.14 | 0.04 | 0.17 | 0.57 | 0.04 | **0.98** | **0.04** | **1.00** | 0.03 |
| DroneRun | 0.08 | 0.66 | 0.00 | 0.00 | **0.91** | **0.75** | 0.30 | 0.04 | 0.69 | 0.08 | 0.55 | 0.32 |
| DroneCircle | 0.10 | 0.68 | **0.78** | **0.40** | 0.22 | 0.68 | 0.03 | 0.03 | 0.50 | 0.27 | **0.99** | **0.51** |
| BulletGym Avg. | 0.15 | 0.55 | 0.55 | 0.29 | 0.33 | 0.58 | 0.30 | 0.20 | **0.79** | 0.27 | **0.89** | **0.39** |

# 5 EXPERIMENTS

## 5.1 DSRL BENCHMARK

We first evaluate AEGIS on the SafetyGymnasium (Ji et al., 2023) and BulletSafetyGym (Gronauer, 2022) task suites of the DSRL benchmark (Liu et al., 2024). The SafetyGymnasium task suites consist of tasks where agents navigate to goals while avoiding dangerous collisions or contact, and tasks involving robot control under velocity constraints. BulletSafetyGym comprises locomotion tasks under physical safety constraints.

**Baselines.** We compare AEGIS with the following methods: 1) *BCQ-Lag*: BCQ (Fujimoto et al., 2019) augmented with a PID-Lagrangian approach (Stooke et al., 2020). 2) *COptiDICE* (Lee et al., 2022): an offline safe RL method based on distribution-correction estimator. The above two methods are based on expectation-based constraints. To enable fair comparison with our almost-sure constraint setting, we also include: 3) $\gamma$-*CDT*: Constrained Decision Transformer (Liu et al., 2023), which samples a trajectories conditioned on reward and cost returns. We adapt it to use $\gamma$-discounted returns, providing remaining budget as the target cost return. 4) *TREBI* (Lin et al., 2023): a diffusion-based planner that samples trajectories while respecting the remaining budget constraint. 5) *IDQL-Sauté*: diffusion-based RL (Hansen-Estruch et al., 2023) on the Sauté MDP (Sootla et al., 2022), which gets reward penalty when remaining budget is negative. To match AEGIS, we guide the diffusion policy training using importance-weight. Further implementation details are provided in Appendix D.

**Metrics.** The primary evaluation criteria are safety and the reward returns under safety constraints. While the discounted cost and reward returns of the episodes quantify these criteria, their simple averages might suffer from lucky trajectories or infeasible-but-high-rewarding trajectories. Therefore, we define two complementary metrics. The *Feasibility Rate* (FR) is the rate of generated trajectories that remain feasible, i.e., $\delta_t \geq 0$ for all $t$. The *Constrained Reward Return* (CRR) is the average discounted reward return computed over feasible trajectories only. We normalize this return by the minimum and maximum return in the offline data.

Table 1 presents quantitative results for an initial threshold $\delta_0 = \frac{1}{(1-\gamma)}\frac{20}{L}$, where $L$ denotes the maximum episode steps and $\gamma$ is the discount factor. We first note that the initial state might not always be feasible from the given initial budget across all tasks, suggesting that achieving a feasibility rate of 1 could be impossible for some tasks.

Both BCQ-Lag and COptiDICE exhibit infeasible trajectories in over 25% of episodes across various tasks in the DSRL benchmark. This underscores the need for methods ensuring almost-sure constraint satisfaction. Moreover, feasibility rates of $\gamma$-CDT and TREBI also fell below 0.75 in several environments. Notably, IDQL-Sauté achieves lower feasibility than AEGIS, despite using a

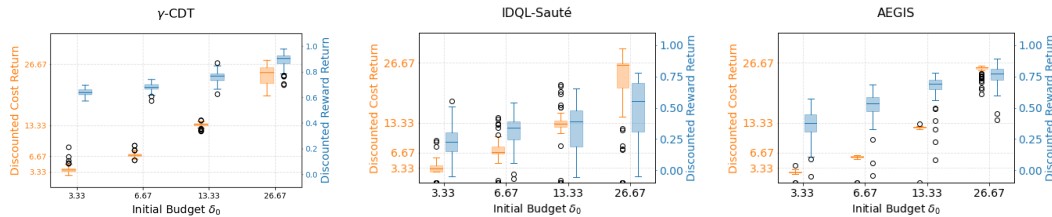

Figure 1: Box plots of discounted cost return (orange) and discounted reward return (blue) with respect to the initial budget $\delta_0$ in the DroneCircle task. Reward is normalized as in constrained reward return.

substantial reward penalty ($n = 100$). AEGIS restricts actions and imposes penalties based on the feasibility Q-function before exhausting the budget, thus enhancing safety.

In contrast, AEGIS maintains feasibility rates exceeding 75% across nine tasks, excluding DroneRun. Although it is possible that hyperparameter tuning specific to tasks yielded feasibility rates above 0.9 even for DroneRun, Table 1 uniformly applies the same hyperparameters across all tasks for fair comparison. Additional experimental results related to this are provided in the Appendix E.1. AEGIS also achieves constrained reward returns comparable to the baseline methods. While AEGIS might exhibit lower constrained reward returns than certain algorithms in specific environments, maintaining high feasibility rates necessitates conservative actions, inherently limiting the constrained reward returns.

Figure 1 illustrates how returns vary with the initial budget $\delta_0$ in the DroneCircle task, without any additional training. Because the reward–cost trade-off shifts with safety budget, the ability to adapt to different budgets is crucial. Unlike the two baselines, AEGIS maintains high feasibility across all four budget settings, indicating robust budget generalization. Overall, on the DSRL benchmark, AEGIS consistently achieves high feasibility with competitive rewards, and adapts to various safety budgets.

## 5.2 HUMANOID LOCOMOTION

To demonstrate the scalability of our method, we further evaluate AEGIS on a high-dimensional humanoid locomotion task using the Unitree G1 (Robotics, 2024). The task is built upon the HumanoidVerse (LeCAR Lab, 2024) framework, requiring the humanoid to move forward while remaining within the energy budget. To this end, we collected a diverse set of trajectories in IsaacGym (Makoviychuk et al., 2021), each labeled with locomotion rewards and energy costs. Figure 2 illustrates the simulation environment and the full specification of the humanoid locomotion task is provided in Appendix D.2.

Figure 3 demonstrates the reward-cost trade-off in the humanoid locomotion task, highlighting the post-training adaptivity of AEGIS. We observe that the resulting behaviors vary according to the initial budget $\delta_0$. Figure 4 illustrates the corresponding changes in average velocity using box plots. In these plots, the orange line denotes the median, the box outlines the interquartile range (IQR), and the whiskers extend to the minimum and maximum data values within 1.5 IQR from Q1 and Q3. Snapshots illustrating the inferred trajectories are in the Appendix E.2.

## 6 RELATED WORK

**Safe RL** Various approaches based on expectation constraints, such as primal-dual (Stooke et al., 2020) and trust-region methods (Achiam et al., 2017; Kim and Oh, 2022), have been proposed. Offline safe RL poses unique challenges due to dataset limitations and a lack of environment interaction. Lee et al. (2022) leverage distribution corrections to achieve conservative expected constraints, yet their work remains focused on expectation-based safety, not ensuring almost-sure guarantees. Some studies introduce risk measures like CVaR to achieve tighter risk control (Kim and Oh, 2022; Yang et al., 2021). While these methods use cost return distributions to manage worst-case scenarios, the inherent multi-step stochasticity of policies complicates their application in offline settings like

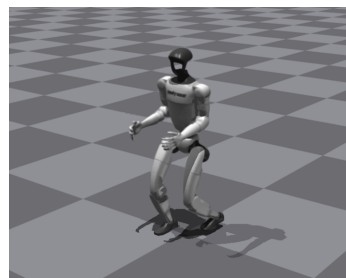

Figure 2: Unitree G1

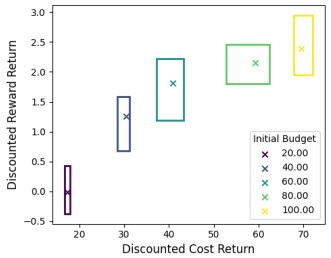

Figure 3: 2D box plot of returns in humanoid locomotion task.

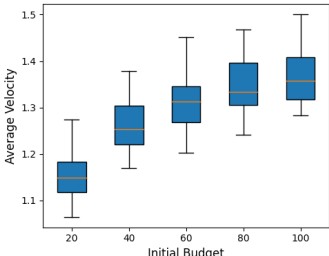

Figure 4: Average velocity in humanoid locomotion task, with respect to the initial budget $\delta_0$

IQL. To ensure almost-sure safety guarantees, Sootla et al. (2022) augmented the state with a safety budget, inspiring our formulation. Yet, most of these works either assume or test exclusively in online interactive settings.

**Diffusion Policy for Offline RL** There have been numerous attempts to apply diffusion models to offline RL, some of which introduce diffusion policies that sample actions conditioned on the current state. By leveraging the ability of diffusion models to accurately capture the behavior policy distribution, these methods can generate diverse and high-quality actions, enabling effective policy extraction from offline data (Wang et al., 2023; Hansen-Estruch et al., 2023). Furthermore, many diffusion-based offline RL methods incorporate value-based guidance during inference (Lu et al., 2023) or training (Zhang et al., 2025) to bias sampling toward high-reward actions. However, because these policies are guided primarily by reward without explicit consideration of safety, they remain safety-agnostic and may replicate risky behaviors embedded in the dataset.

Recent attempts blend diffusion models with offline safe RL. Zheng et al. (2024) mask unsafe diffusion-generated actions based on reachability analysis, achieving strict constraints but potentially overly conservative by not tolerating any budgets. Furthermore, its reward guidance within the feasible region relies on a constraint-agnostic reward critic, which may favor actions that are suboptimal under the true feasible policy. Lin et al. (2023) apply diffusion at the trajectory optimization level, conditioning on budget constraints, but suffers from computational complexity.

## 7 LIMITATION

While AEGIS demonstrates a significant step towards almost surely safe offline reinforcement learning, several limitations exist. First, the almost surely safety guarantee is contingent upon the expectile approximation. Our empirical results reflect that feasibility rates, while high, do not consistently reach 100% or the highest possible rates. Second, AEGIS can be sensitive to hyperparameters such as the expectile levels $\alpha$ and reward penalty $n$, particularly affecting the reward-cost trade-off. Although Table 1 used the same hyperparameters for all tasks for fair comparison, the ablation study in the Appendix E.1 shows room for improvement.

## 8 CONCLUSION

Leveraging diffusion guidance, AEGIS reframes offline safe RL by learning an expectile-based feasibility critic whose epigraph approximately maps the state–budget pairs that enable near almost-sure safety. Coupled with reward critics, this representation lets a single diffusion policy yield actions that are simultaneously high-return and constraint-satisfying across a wide range of user-specified budgets, without additional tuning. Empirically, AEGIS maintains a feasibility above 75% across most DSRL tasks while adaptively maximizing rewards based on the available budget. Additionally, AEGIS learned diverse behaviors according to energy budgets in the humanoid locomotion task, enabling adaptive deployment without post-training. Overall, our feasibility-guided diffusion policy provides a readily deployable safety mechanism for offline RL. A single policy trained once enforces risk-sensitive constraints across those budgets and paves the way for scalable constrained control.

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

## A    USE OF LARGE LANGUAGE MODELS (LLMS)

We used Large Language Models (LLMs) solely to aid or polish the writing of this paper. Specifically, LLMs were employed to improve the grammar, phrasing, and readability of the text, without altering the technical content, claims, or experimental results. The authors take full responsibility for all content, including any text generated with the assistance of an LLM, and confirm that LLM usage did not contribute at a level warranting authorship.

## B    PROOFS

### B.1    PROOF OF PROPOSITION 3.1

Consider the restricted *restricted* MDP whose state space is $\mathcal{F}$ and only viable actions are admissible at each $\tilde{s} \in \mathcal{F}$. To justify this construction, we first show that every $\tilde{s} \in \mathcal{F}$ admits at least one viable action. Suppose that there exist $\tilde{s}_0 \in \mathcal{F}$ with no viable action. By definition of action viability, for every action $a$ we have $P(\tilde{s}_1 \notin \mathcal{F} \mid \tilde{s}_0, a) > 0$. Consequently, under any policy $\pi$, the successor $\tilde{s}_1$ lies outside $\mathcal{F}$ with positive probability. By the definition of the feasible set $\mathcal{F}$, this implies the remaining budget at some timestep becomes negative with nonzero probability, so no policy satisfies the almost-sure constraint from $\tilde{s}_0$. Hence, $s_0 \notin \mathcal{F}$, a contradiction. Therefore, every $\tilde{s} \in \mathcal{F}$ has at least one viable action.

Standard discounted–MDP theory (Puterman, 1994) yields a deterministic stationary optimal policy $\pi^*_{\mathrm{res}}$ for the restricted MDP. Define a deterministic policy $\pi^*$ for the original (augmented) MDP that plays as $\pi^*_{\mathrm{res}}$ on $\mathcal{F}$ and plays an arbitrary action elsewhere. We want show that the $\pi^*$ is a solution to (3) for an arbitrary $\tilde{s}_0 \in \mathcal{F}$.

For any $\tilde{s}_t \in \mathcal{F}$, the policy $\pi^*$ selects a viable action, hence $\tilde{s}_{t+1} \in \mathcal{F}$ almost surely. By induction on $t$, $\pi^*$ satisfies the almost-sure constraint. Next, we show that no policy $\pi$ satisfying the constraint from $\tilde{s}_0$ can achieve a higher reward objective than $\pi^*$. Let $\pi$ be any policy that satisfies the almost-sure constraint from $\tilde{s}_0 \in \mathcal{F}$, and let $\mathcal{I}$ be the set of augmented states that are reachable from $\tilde{s}_0$ with positive probability under $\pi$. If there exists $\tilde{s} \in \mathcal{I}$ such that $a \sim \pi(\cdot \mid \tilde{s})$ assigns positive probability to a non-viable action at $\tilde{s}$, then the constraint would be violated from $\tilde{s}_0$; therefore, on $\mathcal{I}$ the policy $\pi$ selects only viable actions almost surely. Define a policy $\pi_{\mathrm{res}}$ on the restricted MDP that agrees with $\pi$ on $\mathcal{I} \cap \mathcal{F}$ and takes any viable action elsewhere. Since trajectories starting from $\tilde{s}_0 \in \mathcal{F}$ remain in $\mathcal{F}$ almost surely under $\pi$, the reward objective of $\pi$ (in the original MDP) equals that of $\pi_{\mathrm{res}}$ (in the restricted MDP). Likewise, the reward objective of $\pi^*$ equals that of $\pi^*_{\mathrm{res}}$. Because $\pi^*_{\mathrm{res}}$ is optimal for the restricted MDP, $\pi$ cannot achieve a higher reward than $\pi^*$ from $\tilde{s}_0$. Hence, $\pi^*$ solves (3) for any initial $\tilde{s}_0 \in \mathcal{F}$.

### B.2    PROOF OF PROPOSITION 3.2

Recall that an augmented state $\tilde{s} = (s, \delta) \in \tilde{\mathcal{S}}$ belongs to the feasible set $\mathcal{F}$ if and only if there exists a policy that almost surely keeps the remaining budget non-negative for all time. Define $\mathcal{K}$ as the set of augmented states at which there exists a viable action. Our first goal is to show that $\mathcal{F} = \mathcal{K}$.

Let $\tilde{s}_0 \notin \mathcal{K}$. By definition of $\mathcal{K}$, for *every* action $a$ we have $P(\tilde{s}_1 \notin \mathcal{F} \mid \tilde{s}_0, a) > 0$. Consequently, for *any* policy $\pi$ the probability that the successor $\tilde{s}_1$ lies outside $\mathcal{F}$ is strictly positive, and the remaining budget becomes negative with nonzero probability. Hence no policy can satisfy the almost-sure constraint from $\tilde{s}_0$, so $\tilde{s}_0 \notin \mathcal{F}$. Therefore $\mathcal{F} \subset \mathcal{K}$.

Define the regulation map $\mathcal{R}$ that assigns to each $\tilde{s} \in \mathcal{K}$ the set of actions viable at $\tilde{s}$. By the measurable-selection theorem (Kuratowski and Ryll-Nardzewski, 1965), there exists a measurable selector $f : \mathcal{K} \to \mathcal{A}$ with $f(\tilde{s}) \in \mathcal{R}(\tilde{s})$ for all $\tilde{s} \in \mathcal{K}$. Consider a deterministic policy $\pi_0$ that plays $f(\tilde{s})$ at $\tilde{s} \in \mathcal{K}$ and some arbitrary $a_0$ elsewhere. For any $\tilde{s}_t \in \mathcal{K}$, $\delta_t \geq 0$ and $\tilde{s}_{t+1} \in \mathcal{F} \subset \mathcal{K}$ almost-surely with this policy. Thus every $\tilde{s}_0 \in \mathcal{K}$ belongs to $\mathcal{F}$, i.e. $\mathcal{K} \subset \mathcal{F}$.

Combining both inclusions yields $\mathcal{F} = \mathcal{K}$. We now address the policy part of the theorem. Suppose a policy $\pi$ is such that, for every $\tilde{s} \in \mathcal{F}$, $a \sim \pi(\cdot|\tilde{s})$ is viable at $\tilde{s}$ almost surely. Then, whenever $s_t \in \mathcal{F}$, we have $s_{t+1} \in \mathcal{F}$ almost surely; in particular, if $s_0 \in \mathcal{F}$, the trajectory stays in $\mathcal{F}$ almost surely. Thus $\pi$ is feasible on $\mathcal{F}$. Conversely, if there exists $\tilde{s} \in \mathcal{F}$ such that $a \sim \pi(\cdot|\tilde{s})$ is not viable at

$\tilde{s}$ with positive probability, then the policy is not feasible from $s_0 = s$ because $s_1 \notin \mathcal{F}$ with nonzero probability.

## B.3 DERIVATION FROM (5) TO (6)

Recall that the feasible set $F$ is the epigraph of $V_f$. The condition $(s', \delta') \in F$ is equivalent to $\delta' \geq V_f(s')$. Using the budget transition rule $\delta' = \min\left(\frac{\delta - C(s,a)}{\gamma}, \frac{1}{1-\gamma}\right)$ and the fact that $V_f(s') \leq \frac{1}{1-\gamma}$, the condition is also equivalent to $\frac{\delta - C(s,a)}{\gamma} \geq V_f(s')$. Hence, the definition (5) rewrites as

$$Q_f(s,a) = \inf\{\delta | (s', \delta') \in \mathcal{F} \quad \text{a.s. for } (s', \delta') \sim \tilde{\mathcal{P}}(\cdot | (s, \delta), a)\}, \tag{25}$$

$$= \inf\{\delta | \delta \geq C(s,a) + \gamma V_f(s') \quad \text{a.s. for } s' \sim \mathcal{P}(\cdot | s, a)\}, \tag{26}$$

which is equation (6) from the definition of essential supremum.

## B.4 PROOF OF THEOREM 3.3

Feasibility Q-function $Q_f$ is bounded on $[0, \frac{1}{1-\gamma}]$. We first show that the feasibility Bellman operator $\mathcal{T}_f$ is a $\gamma$-contraction under the sup norm. For any $Q_1, Q_2 \colon \mathcal{S} \times \mathcal{A} \to [0, \frac{1}{1-\gamma}]$, define $V_1(s) = \min_a Q_1(s,a)$ and $V_2(s) = \min_a Q_2(s,a)$. Then,

$$\|\mathcal{T}_f Q_1 - \mathcal{T}_f Q_2\|_\infty \tag{27}$$

$$= \sup_{s \in \mathcal{S}, a \in \mathcal{A}} \left| \operatorname*{ess\,sup}_{s' \sim \mathcal{P}(\cdot|s,a)} [C(s,a) + \gamma V_1(s')] - \operatorname*{ess\,sup}_{s' \sim \mathcal{P}(\cdot|s,a)} [C(s,a) + \gamma V_2(s')] \right| \tag{28}$$

$$\leq \sup_{s \in \mathcal{S}, a \in \mathcal{A}} \sup_{s' \in \mathcal{S}} |\{C(s,a) + \gamma V_1(s')\} - \{C(s,a) + \gamma V_2(s')\}| \tag{29}$$

$$= \gamma \sup_{s' \in \mathcal{S}} |V_1(s') - V_2(s')| \tag{30}$$

$$\leq \gamma \sup_{s' \in \mathcal{S}} \sup_{a' \in \mathcal{A}} |Q_1(s', a') - Q_2(s', a')| \tag{31}$$

$$\leq \gamma \|Q_1 - Q_2\|_\infty \tag{32}$$

so $\mathcal{T}_f$ is a $\gamma$-contraction. By the Banach fixed-point theorem (Puterman, 1994), there exists a unique fixed point $Q^*$, and the iteration $Q_{j+1} = \mathcal{T}_f Q_j$ converges to $Q^*$ from any initial bounded $Q_0$. Finally, from equation. (6), the feasibility Q-function $Q_f$ is the fixed-point: $Q_f = \mathcal{T}_f Q_f$. Hence, $Q^* = Q_f$.

## B.5 CONVERGENCE OF $Q_f^\alpha$ TO $Q_f$

Similarly to Appendix B.4, the expectile-based operator in (9) is a $\gamma$-contraction under the sup norm and converges to a unique fixed point. Here we study the fixed point $Q_f^\alpha$. Consider the sequence of functions defined by $Q_0 = Q_f$, $Q_{i+1} = \mathcal{T}_f^\alpha Q_i$. For any $s \in \mathcal{S}$ and $a \in \mathcal{A}$,

$$|Q_{i+1}(s,a) - Q_f(s,a)| \leq \left| \mathcal{T}_f^\alpha Q_i(x) - \mathcal{T}_f^\alpha Q_f(x) \right| + \left| \mathcal{T}_f^\alpha Q_f(s,a) - Q_f(s,a) \right|, \tag{33}$$

$$\leq \gamma |Q_i(s,a) - Q_f(s,a)| + \left| \mathcal{T}_f^\alpha Q_f(x) - Q_f(x) \right|. \tag{34}$$

By induction, $|Q_i(x) - Q_f(x)| \leq \frac{1-\gamma^i}{1-\gamma} |\mathcal{T}_f^\alpha Q_f(x) - Q_f(x)| \leq \frac{1}{1-\gamma} |\mathcal{T}_f^\alpha Q_f(x) - Q_f(x)|$ holds for any $i \in \mathbb{Z}_0^+$. Here, $\mathcal{T}_f^\alpha Q_f(s,a)$ converges to $Q_f(s,a) = \mathcal{T}_f Q_f(s,a)$ by the relationship between the expectile and the essential supremum. Hence, as $\alpha \to 1$, $\mathcal{T}_f^\alpha Q_f(s,a)$ converges to $Q_f(s,a)$.

## B.6 PRECISE STATEMENT OF THEOREM 4.1 AND ITS PROOF

**Theorem B.1** (Precise version of Theorem 4.1). *Let $\pi^*$ be a policy that is a common solution to the constrained optimization problem (3) for all initial state–budget pairs $(s_0, \delta_0)$ in the feasible set $\mathcal{F}$. Then, for any $(s_0, \delta_0) \in \mathcal{F}$, the following parts hold:*

1. *For any policy $\pi$ that is not feasible, we have:*

    (a) *There exists $\alpha_0 \in (0, 1)$ such that for all $\alpha \in [\alpha_0, 1)$, the policy $\pi$ is not a solution to (11).*

    (b) *There exists $n_0 \in \mathbb{R}_0^+$ such that for all $n \geq n_0$, the policy $\pi$ is not a solution to (11).*

2. *Among feasible policies, $\pi^*$ maximizes the objective value in (11). Moreover, $\pi^*$ satisfies the constraint in (11).*

Part 1 states that any infeasible policy cannot be a solution to (11) in the limits $\alpha \to 1$ or $n \to \infty$. Part 2 then says that among the remaining (feasible) policies, $\pi^*$ both maximizes the objective in (11) and satisfies its constraint. Together, these two parts show that the precise theorem recovers the content of the informal Theorem 4.1.

*Proof.*

**Proof of Part 1a.** Assume $\pi$ is not feasible. Then, under trajectories induced by $\pi$, there is a set of augmented states $(s, \delta) \in \mathcal{F}$ and actions $a \sim \pi(\cdot|s, \delta)$ with positive probability on which $Q_f(s, a) > \delta$ (feasibility violation). By convergence $Q_f^\alpha \to Q_f$ and the monotonicity , there exists $\alpha_0 \in (0, 1)$ such that for all $\alpha \in [\alpha_0, 1)$ we also have $Q_f^\alpha(s, a) > \delta$ on that set with positive probability. Hence $\pi$ violates the constraint of (11) and cannot be a solution, proving Part 1a.

**Proof of Part 1b.** Suppose $\pi$ is not feasible. Then with positive probability along trajectories under $\pi$ there exists a time $t$ such that $V_f(s_t) > \delta_t$. Since $V_f^\alpha \leq V_f$ , the same event implies $V_f^\alpha(s_t) > \delta_t$. By the definition of the augmented reward $\tilde{R}$ in (11), incurring such a violation triggers a penalty of size $n$. Let $p > 0$ be the probability of at least one violation under $\pi$. The expected total penalty contribution is then at least $p\, n$, which drives the objective in (11) arbitrarily low as $n \to \infty$. Hence there exists $n_0$ such that for all $n \geq n_0$, the value achieved by $\pi$ is strictly worse than that of some feasible policy (e.g., $\pi^*$), so $\pi$ cannot be optimal. This proves Part 1b.

**Proof of Part 2.** Fix any feasible policy $\pi$ and any $\tilde{s}_0 \in \mathcal{F}$. Feasibility means that along trajectories we have $V_f(s_t) \leq \delta_t$ almost surely, hence also $V_f^\alpha(s_t) \leq \delta_t$ almost surely. Therefore the augmented reward $\tilde{R}(s_t, \delta_t, a_t)$ equals the original reward $R(s_t, a_t)$ almost surely, so the objective in (11) coincides with the reward objective of (3) when restricted to feasible policies. Since $\pi^*$ maximizes the latter among feasible policies by assumption, it also maximizes the objective of (11) among feasible policies.

Finally, because $\pi^*$ is feasible on $\mathcal{F}$, for any $(s, \delta) \in \mathcal{F}$ and $a \sim \pi^*(\cdot|s, \delta)$ we have $Q_f(s, a) \leq \delta$ almost surely. Using $Q_f^\alpha \leq Q_f$ yields $Q_f^\alpha(s, a) \leq \delta$ almost surely, so $\pi^*$ satisfies the constraint in (11). This completes the proof of Part 2. $\qquad\square$

## C  FEASIBILITY Q-FUNCTION ON A SIMPLE ENVIRONMENT

To verify the behavior of the feasibility Q-function and its expectile approximation in the simplest setting, we consider a two-state, single-action MDP. The state space is $\{H, T\}$ and the next state is drawn randomly with probability $\mathcal{T}(s' = H|s, a) = 1 - p$ and $\mathcal{T}(s' = T|s, a) = p$. The cost function $C$ is given by $C(H, a) = 0$ and $C(T, a) = 1$.

Figure 5 visualizes the expectile approximation $Q_f^\alpha$ against the expectile level $\alpha$. When $\alpha = 0.5$, the update reduces to the expectation used by the standard Bellman operator, yielding a smaller value than the feasibility Q-function. As $\alpha$ approaches 1, $Q_f^\alpha$ increases and converges to the feasibility Q-function.

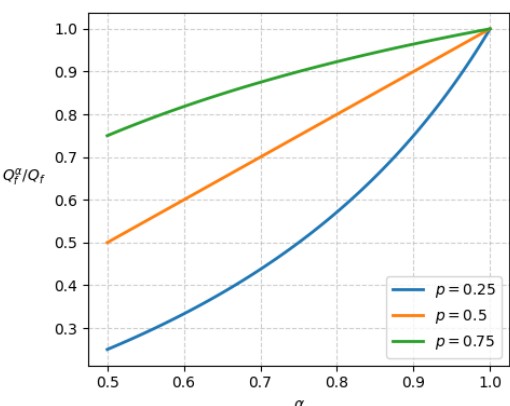

Figure 5: $Q_f^\alpha / Q_f$ on the two-state single-action environment.

## D  IMPLEMENTATION DETAILS

### D.1  ALGORITHM IMPLEMENTATION AND HYPERPARAMETERS

We run BCQ-Lag, COptiDICE, and CDT ($\gamma$-CDT) baselines using the implementations and hyperparameters provided by the benchmark suite[1] (Liu et al., 2024). For $\gamma$-CDT, we modify the CDT implementation to take *discounted* cost returns, with discount factor $\gamma = 0.99$, instead of undiscounted one as input. TREBI is evaluated with the official implementation and hyperparameters[2].

IDQL-Sauté and our proposed AEGIS are implemented on top of the Zheng et al. (2024) codebase[3]. Both the feasibility critics and the reward critics are three layer MLPs with a hidden dimension of 256. The diffusion model follows the IDQL (Hansen-Estruch et al., 2023) architecture. To condition on the safety budget when generating actions, we concatenate the budget encoding with the noised action, state, and time encoding at the input. At inference, the diffusion model samples $N$ candidate actions and then selects a single action as follows:

- **IDQL-Sauté:** The action with the highest (reward) Q-value.

- **AEGIS:** The action with the highest reward Q-value, among those whose feasibility critic is within the budget. If no sampled action is feasible, select the action with the smallest feasibility Q-value.

The hyperparameters used for IDQL-Sauté and AEGIS are listed Table 2.

---

[1] https://github.com/liuzuxin/OSRL

[2] https://github.com/qianlin04/Safe-offline-RL-with-diffusion-model

[3] https://github.com/ZhengYinan-AIR/FISOR

Table 2: Hyperparameters of IDQL-Sauté and AEGIS

| Parameter | Value |
|---|---|
| Discount factor $\gamma$ | 0.99 |
| Training steps | 1e6 |
| Learning rate | 3e-4 |
| Optimizer | Adam |
| Activation function | ReLU |
| Critic batch size | 256 |
| Actor batch size | 2048 |
| Reward penalty $n$ | 100 (IDQL-Sauté), 8 (AEGIS) |
| Expectile level $\alpha$ for operator | 0.9 |
| Expectile level $\tau_f$ for implicit feasibility value learning | 0.9 |
| Expectile level $\tau_r$ for implicit reward value learning | 0.9 |
| Actor temperature $\beta$ | 3 |
| Diffusion step $T$ | 5 |
| Number of action candidates $N$ | 256 |

Table 3: Reward specifications for humanoid locomotion. **Notation:** $v_x, v_y$: base linear velocities; $\omega_z$: base yaw angular velocity; $h_z$: base height; $\hat{h}_z$: target base height; $\mathbf{v}_{\text{foot}}$: foot velocity; $q_j$: joint angle; $t_{\text{air}}$: elapsed air time since last liftoff; $\theta$: heading angle; $\mathbf{g}$: gravity vector; $h_{\text{foot}}$: foot height; $\hat{h}_{\text{foot}}$: target foot height; $\mathbf{p}$: position vector; $F_{\text{foot}}^z$: vertical contact force at the foot.

| Term | Expression | Weight |
|---|---|---|
| Forward velocity | $\lvert v_x \rvert^2$ | 2.0 |
| Angular velocity penalty | $\exp(-\lvert \omega_z \rvert^2 / 0.25)$ | 0.5 |
| Base height deviation | $\lvert h_z - \hat{h}_z \rvert^2$ | -10.0 |
| Termination penalty | $\mathbb{1}\{termination\}$ | -1.0 |
| Hip yaw & roll penalty | $\sum_{j \in \text{hip yaw \& roll joints}} \lvert q_j \rvert^2$ | -1.0 |
| Feet heading alignment | $\lvert \theta_{\text{left foot}} - \theta_z \rvert + \lvert \theta_{\text{right foot}} - \theta_z \rvert$ | -0.3 |
| Feet orientation | $\lVert \mathbf{g} - \mathbf{g}_{\text{feet}}^{proj} \rVert^2$ | -1.0 |
| Feet away | $\mathbb{1}\{\lVert \mathbf{p}_{\text{left foot}} - \mathbf{p}_{\text{right foot}} \rVert^2 < 0.15\}$ | -2.0 |
| Feet air time | $\sum_{\text{foot}} \mathbb{1}\{first\ contact\} \cdot \max(t_{\text{air}} - 0.5,\ 0.0)$ | 1.0 |
| Feet height | $\sum_{\text{foot}} \max(\lvert h_{\text{foot}} - \hat{h}_{\text{foot}} \rvert^2 - 0.02, 0)$ | -5.0 |
| Slippage penalty | $\sum_{\text{foot}} \mathbb{1}\{F_{\text{foot} z} \geq 1.0\} \cdot \lVert \mathbf{v}_{\text{foot}} \rVert^2$ | -1.0 |
| Energy cost | $\lvert v_x \rvert^2 + \lvert v_y \rvert^2$ | - |

## D.2 HUMANOID LOCOMOTION TASK

Based on the HumanoidVerse framework (LeCAR Lab, 2024), we constructed the humanoid loco-motion task for forward walking. The observation space, with a total dimension of 252, comprises base angular velocity, heading vector, past actions, joint positions, velocities, and their histories. The action space corresponds to the target joint positions, with a dimension of 12. The reward components and their associated weights are summarized in Table 3. In particular, the velocity reward was modified from the original HumanoidVerse implementation to account only for the forward direction. Furthermore, the energy cost term was defined as the squared velocity on the $xy$-plane, representing the energetic expenditure of the humanoid during locomotion.

Table 4: Results for the feasibility rate (FR) and constrained reward return (CRR) by expectile level $\alpha$. Higher is better. Each value is averaged over 50 evaluation episodes and three distinct random seeds.

| Expectile level $\alpha$ | 0.5 | | 0.7 | | 0.9 | | 0.95 | |
|---|---|---|---|---|---|---|---|---|
| | FR ↑ | CRR ↑ | FR ↑ | CRR ↑ | FR ↑ | CRR ↑ | FR ↑ | CRR ↑ |
| CarButton1 | 0.24 | 0.28 | 0.43 | 0.24 | 0.79 | 0.10 | 0.73 | 0.10 |
| CarGoal1 | 0.65 | 0.61 | 0.64 | 0.48 | 0.95 | 0.18 | 0.97 | 0.14 |
| PointButton1 | 0.43 | 0.30 | 0.68 | 0.22 | 0.80 | 0.12 | 0.73 | 0.11 |
| PointGoal1 | 0.56 | 0.60 | 0.75 | 0.24 | 0.75 | 0.18 | 0.81 | 0.10 |
| HalfCheetah Vel | 1.00 | 0.96 | 1.00 | 0.95 | 1.00 | 0.84 | 1.00 | 0.67 |
| Swimmer Vel | 0.43 | 0.20 | 0.43 | 0.21 | 0.91 | 0.20 | 0.93 | 0.21 |
| SafetyGym Avg. | 0.55 | 0.49 | 0.65 | 0.39 | 0.87 | 0.27 | 0.86 | 0.22 |
| AntRun | 1.00 | 0.70 | 1.00 | 0.68 | 1.00 | 0.71 | 1.00 | 0.69 |
| AntCircle | 0.96 | 0.53 | 0.99 | 0.36 | 1.00 | 0.03 | 1.00 | 0.01 |
| DroneRun | 0.75 | 0.67 | 0.50 | 0.67 | 0.55 | 0.32 | 1.00 | 0.10 |
| DroneCircle | 0.88 | 0.56 | 0.95 | 0.55 | 0.99 | 0.51 | 1.00 | 0.42 |
| BulletGym Avg. | 0.90 | 0.61 | 0.86 | 0.56 | 0.89 | 0.39 | 1.00 | 0.31 |

Table 5: Results for the feasibility rate (FR) and constrained reward return (CRR) by reward penalty $n$. Higher is better. Each value is averaged over 50 evaluation episodes and three distinct random seeds.

| Reward penalty $n$ | 0.0 | | 4.0 | | 8.0 | | 12.0 | |
|---|---|---|---|---|---|---|---|---|
| | FR ↑ | CRR ↑ | FR ↑ | CRR ↑ | FR ↑ | CRR ↑ | FR ↑ | CRR ↑ |
| CarButton1 | 0.74 | 0.15 | 0.79 | 0.12 | 0.79 | 0.10 | 0.80 | 0.11 |
| CarGoal1 | 0.89 | 0.35 | 0.95 | 0.21 | 0.95 | 0.18 | 0.93 | 0.13 |
| PointButton1 | 0.69 | 0.18 | 0.79 | 0.13 | 0.80 | 0.12 | 0.78 | 0.11 |
| PointGoal1 | 0.67 | 0.27 | 0.68 | 0.23 | 0.75 | 0.18 | 0.73 | 0.16 |
| HalfCheetah Vel | 1.00 | 0.97 | 1.00 | 0.89 | 1.00 | 0.84 | 1.00 | 0.82 |
| Swimmer Vel | 0.58 | 0.16 | 0.87 | 0.17 | 0.91 | 0.20 | 0.90 | 0.26 |
| SafetyGym Avg. | 0.76 | 0.35 | 0.85 | 0.29 | 0.87 | 0.27 | 0.86 | 0.26 |
| AntRun | 0.83 | 0.68 | 0.93 | 0.71 | 1.00 | 0.71 | 1.00 | 0.71 |
| AntCircle | 1.00 | 0.03 | 1.00 | 0.03 | 1.00 | 0.03 | 1.00 | 0.02 |
| DroneRun | 0.44 | 0.57 | 0.27 | 0.38 | 0.55 | 0.32 | 0.65 | 0.38 |
| DroneCircle | 0.35 | 0.63 | 0.90 | 0.56 | 0.99 | 0.51 | 0.99 | 0.43 |
| BulletGym Avg. | 0.66 | 0.48 | 0.77 | 0.42 | 0.89 | 0.39 | 0.91 | 0.38 |

# E ADDITIONAL RESULTS

## E.1 ABLATION STUDY AND HYPERPARAMETER ANALYSIS

Table 4 examines the effect of using worst-case backup and its expectile level, in the feasibility Bellman operator. Relative to the standard expectation at $\alpha = 0.5$, increasing $\alpha$ generally raises the feasibility rate, indicating that the expectile backup plays a key role in improving safety. At the same time, higher $\alpha$ tends to reduce the constrained reward return, revealing a clear safety–performance trade-off.

Table 5 studies the reward penalty and its scale $n$. Even with $n = 0$, our method achieves a higher feasibility rate than all baselines except IDQL-Sauté; adding a modest penalty further increases the probability of satisfying safety. A penalty scale of $n = 8$ already yields strong performance, and increasing to $n = 12$ provides only marginal additional gains.

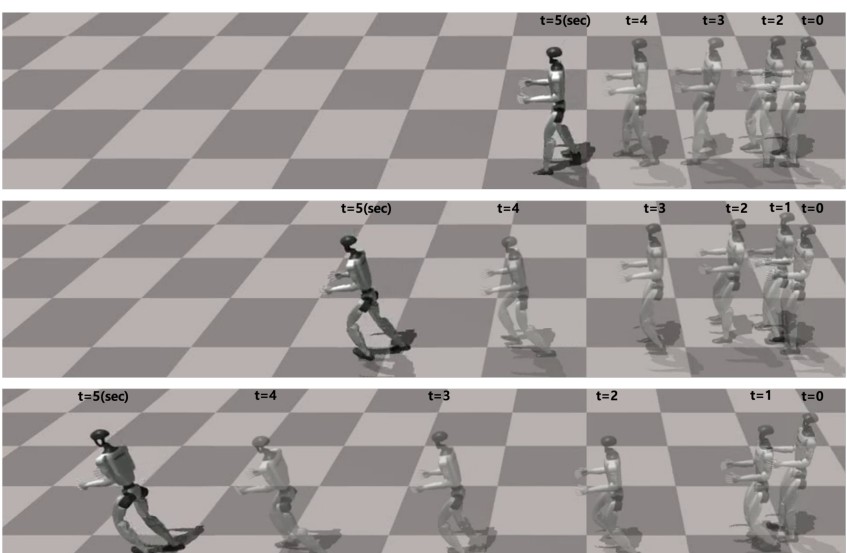

Figure 6: Snapshots of humanoid forward locomotion under different energy budgets, where the top row corresponds to a budget of 20, the middle row to a budget of 40, and the bottom row to a budget of 80. Each sequence illustrates a 5-second rollout.

### E.2 Qualitative Results on Humanoid Locomotion

Fig. 6 shows humanoid locomotion snapshots under initial budgets of 20, 40, and 80. When the initial budget is set to 20, the humanoid requires a relatively long period to reach a sufficient forward velocity, since a substantial amount of energy is needed for the initially stationary agent to accelerate from rest. In contrast, with a larger initial budget of 80, the humanoid rapidly attains its peak velocity from the very beginning, exhibiting a strong burst of forward movement. The intermediate case of budget 40 lies between these two extremes, showing a moderate locomotion profile. These observations show that AEGIS consistently achieves stable forward locomotion across different initial budgets, thereby demonstrating both robustness and adaptability.

