# OpenReview forum: "AEGIS: Almost Surely Safe Offline Reinforcement Learning"
_ICLR.cc/2026/Conference — Submitted to ICLR 2026_

### Official Review · Reviewer_CpiR · 2025-10-27

**Soundness:** 2
**Presentation:** 3
**Contribution:** 1
**Rating:** 2
**Confidence:** 4

**Summary:**

This paper studies the problem of learning an almost surely safe policy that can be deployed to tasks with varying safety budgets from offline data. The authors propose an framework called AEGIS that augments the state with the remaining safety budget to deal with varying budgets, and learns a worst-case feasibility critic to ensure almost safety. The algorithm learns a diffusion policy guided by both feasibility and reward critics. Experimental results on DSRL benchmark and humanoid locomotion tasks show that AEGIS achieves high feasibility with competitive returns.

**Strengths:**

1. The proposed methods are theoretically sound, supported by theorems with proofs.
2. The writing of the paper is clear.

**Weaknesses:**

1. The contribution of this paper is unclear. This claimed contribution is learning a policy from offline data that (1) is almost surely safe, and (2) can handle varying budgets. However, learning an almost surely safe policy is already achieved by Sootla et al. [1], and methods for handling varying budgets are already proposed by Lee et al. [2] and Guo et al. [3].
2. The novelty of this paper is unclear. Many of the proposed concepts and methods are similar to existing works. The feasible set defined in Section 3.1 is similar to the concept of maximum feasible region in safe control and safe RL (see [4]). The feasibility value function defined in Eq. (4) is similar to the optimal cost value function in standard safe RL. The feasibility Bellman operator defined in Section 3.3 is essentially a robust Bellman operator (see [5]). The expectile regression for value function learning is a widely used technique in offline RL [6].
3. The necessity of the reward penalty is a flaw of the proposed algorithm. This penalty should be sufficiently large to ensure safety, which seems difficult to choose in practice.

[1] Sootla, A., Cowen-Rivers, A. I., Jafferjee, T., Wang, Z., Mguni, D. H., Wang, J., & Ammar, H. (2022, June). Sauté rl: Almost surely safe reinforcement learning using state augmentation. In *International Conference on Machine Learning* (pp. 20423-20443). PMLR.

[2] Lee, J., Heo, J., Kim, D., Lee, G., & Oh, S. (2023, October). Dual variable actor-critic for adaptive safe reinforcement learning. In *2023 IEEE/RSJ International Conference on Intelligent Robots and Systems (IROS)* (pp. 7568-7573). IEEE.

[3] Guo, Z., Zhou, W., Wang, S., & Li, W. (2025, March). Constraint-conditioned actor-critic for offline safe reinforcement learning. In *The Thirteenth International Conference on Learning Representations*.

[4] Yang, Y., Zheng, Z., Li, S. E., Duan, J., Liu, J., Zhan, X., & Zhang, Y. Q. (2023). Feasible policy iteration. *arXiv preprint arXiv:2304.08845*.

[5] Li, Z., Hu, C., Wang, Y., Yang, Y., & Li, S. E. (2024). Safe reinforcement learning with dual robustness. *IEEE Transactions on Pattern Analysis and Machine Intelligence*.

[6] Kostrikov, I., Nair, A., & Levine, S. (2021). Offline reinforcement learning with implicit q-learning. *arXiv preprint arXiv:2110.06169*.

**Questions:**

1. What is the contribution of this paper with regard to existing works, e.g., [1-3]?
2. What is the novelty of the proposed methods (as detailed in Weakness 2) with regard to existing works?
3. How to choose the reward penalty in practice, and what is its impact on the performance?

---

> ### Author Response · Authors · 2025-11-25
>
> Thank you for your review. We address your points below.
>
> ## Weakness 1 and Question 1
> Our main contributions are as follows:
> - We investigate the feasibility of almost surely safe RL, introducing feasibility critics whose epigraph characterizes the feasible set of state-budget pairs.
> - We present AEGIS, a diffusion-based offline safe RL framework that samples actions respecting risk-sensitive safety across a wide range of safety budgets.
> - Empirically, AEGIS improves the feasibility rate on the DSRL tasks and demonstrates adaptive behavior in the humanoid locomotion task.
>
> 1. Sootla et al. [1] propose almost surely safe RL and a practical algorithm, but they do not characterize which initial augmented states are feasible. In contrast, we theoretically investigate which state–budget pairs are feasible and then design a novel offline algorithm based on this analysis. As discussed around line 249 and 391, AEGIS uses a worst-case, forward-looking feasibility critic to restrict and penalize unsafe actions before the budget is exhausted, while Sootla et al. penalize only when the current remaining budget becomes negative. Empirically, AEGIS outperforms a Sauté-IDQL baseline, which applies Sootla et al. on top of a diffusion-based RL method (IDQL), on the DSRL benchmark.
>
> 2. Lee et al. [2] and Guo et al. [3] handle varying budgets, but as discussed in the related work section, they are based on expectation constraints and cannot be directly applied to almost surely safe RL. Our formulation explicitly targets almost-sure constraints and characterizes feasibility via a worst-case backup in the budget-augmented state space, and our algorithm is designed for the offline setting.
>
> ## Weakness 2 and Question 2
> 1. Feasible sets are indeed of fundamental interest broadly in constrained optimization, including safe control and safe RL. While we agree that feasibility has been studied in various constrained formulations, we believe that the specific analysis we provide for almost surely safe RL with an explicit budget [1] is distinct and novel. Specifically, Yang et al. [4] study a feasible region in the state space without a budget variable, so the feasible set they analyze is structurally different from ours. As stated in Section 2.1 and 3, we focus on feasibility in almost surely safe RL with an explicit budget [1] and show that, in this setting, the feasible set of augmented states can be represented as the epigraph of a feasibility value function.
>
> 2. As described in Section 3.3, the feasibility Q-function resembles the standard Q-function, but it uses the essential supremum over the transition dynamics. This operator captures the worst-case next-state cost rather than “lucky” transitions, ensuring that feasibility is preserved almost surely. We also show that the feasible set is the epigraph of this critic and that we can identify feasible actions via the condition $(Q_f(s,a) \le \delta)$. This is conceptually different from the optimal cost value function in standard safe RL, which directly accumulates costs under a policy and does not explicitly represent the minimal feasible budget for each state.
>
> 3. Our feasibility Bellman operator and robust Bellman operators are similar in that they both use a worst-case backup, but their roles and constructions are different. Robust Bellman operators typically take the worst case over possible environment dynamics (e.g., an uncertainty set of transition kernels [7] or adversarial actions [5]), whereas our operator considers the worst case over stochasticity under a fixed dynamics. As stated in Section 3.3, this yields a policy-agnostic feasibility critic for almost-sure constraints, rather than a robust value function in the usual sense.
>
> 4. As we cite in the paper, using expectile regression for value function learning was introduced by Kostrikov et al. [6]. Building on this idea, we extend expectile-based learning to the almost surely safe RL setting. We note that using expectiles over the dynamics backup for feasibility is not present in [6].

---

> > ### Author Response · Authors · 2025-11-25
> >
> > ## Weakness 3 and Question 3
> > As mentioned around line 395 and in the limitation section, we use a single set of AEGIS hyperparameters uniformly across all tasks in Table 1. Empirically, we found that $n=8.0$ works reliably across all environments we evaluated. We also provide a sensitivity ablation on $n$ in Appendix E.1., Table 5. As expected, increasing $n$ tends to increase the feasibility rate (FR) and decrease the constrained reward return (CRR). Importantly, the method is **not overly sensitive**: $n=4.0$ and $n=12.0$ also achieved higher feasibility rates than the baselines, while maintaining reasonable CRR.
> >
> > ---
> >
> > [1] Sootla, A., Cowen-Rivers, A. I., Jafferjee, T., Wang, Z., Mguni, D. H., Wang, J., & Ammar, H. (2022, June). Sauté rl: Almost surely safe reinforcement learning using state augmentation. In International Conference on Machine Learning (pp. 20423-20443). PMLR.
> >
> > [2] Lee, J., Heo, J., Kim, D., Lee, G., & Oh, S. (2023, October). Dual variable actor-critic for adaptive safe reinforcement learning. In 2023 IEEE/RSJ International Conference on Intelligent Robots and Systems (IROS) (pp. 7568-7573). IEEE.
> >
> > [3] Guo, Z., Zhou, W., Wang, S., & Li, W. (2025, March). Constraint-conditioned actor-critic for offline safe reinforcement learning. In The Thirteenth International Conference on Learning Representations.
> >
> > [4] Yang, Y., Zheng, Z., Li, S. E., Duan, J., Liu, J., Zhan, X., & Zhang, Y. Q. (2023). Feasible policy iteration. arXiv preprint arXiv:2304.08845.
> >
> > [5] Li, Z., Hu, C., Wang, Y., Yang, Y., & Li, S. E. (2024). Safe reinforcement learning with dual robustness. IEEE Transactions on Pattern Analysis and Machine Intelligence.
> >
> > [6] Kostrikov, I., Nair, A., & Levine, S. (2021). Offline reinforcement learning with implicit q-learning. arXiv preprint arXiv:2110.06169.
> >
> > [7] Wang, Y., Velasquez, A., Atia, G., Prater-Bennette, A., & Zou, S. (2023, February). Robust average-reward Markov decision processes. In AAAI Conference on Artificial Intelligence.

---

> ### Comment · Reviewer_CpiR · 2025-11-27
>
> I appreciate the authors' response. However, my primary concerns regarding the paper's novelty and core contribution remain unaddressed. While the proposed method integrates these components in a new way, its foundational elements appear to be adaptations of existing works. Specifically, the almost-sure safety formulation and state augmentation technique are central contributions of Sootla et al. [1]. The characterization of the feasibility set, while presented in a different context, aligns conceptually with well-established ideas in the safe control and safe RL literature. The proposed feasibility Bellman operator, which accounts for stochasticity under a fixed dynamics, also bears a strong conceptual resemblance to the treatment of multiple possible dynamics in robust RL. Consequently, the paper's advance over existing works is insufficient to meet the bar for a novel contribution.

---

### Official Review · Reviewer_rtx1 · 2025-10-30

**Soundness:** 2
**Presentation:** 2
**Contribution:** 3
**Rating:** 4
**Confidence:** 5

**Summary:**

This paper proposes Almost-Sure Epigraph-Guided Implicit Safety (AEGIS), an almost surely safe offline RL framework based on Sauté RL. AEGIS considers both state and cost budget as the epigraph and designs a feasibility critic based on feasibility bellman operator. To approximate estimate the worst-case backup, AEGIS applies expectile regression inspired by IQL. Finally, a diffusion model guided by the feasibility critic is trained as a generative policy to generate high-value feasible actions for various budgets. AEGIS achieves better feasibility rate and constrained reward return compared to some baselines on some tasks.

**Strengths:**

This paper guarantees almost surely safety guarantee on discounted constraints.

This paper enables this guarantee across all feasible budgets through the framework of Sauté RL

The use of IQL to approximate feasibility Bellman operator is well-motivated.

This paper tests their method on a high-dimensional humanoid locomotion task, which is closer to real world cases.

**Weaknesses:**

### Weakness on problem setup

1. Why do you consider the discounted value as the original constraint, considering that the discount factor is just used as an approximation of non-discounted value to ensure the convergence of RL?
   1. Sauté -RL [1] formulates in this way while Paper [2] on Sauté -RL regards the non-discounted value as the original constraint in Sauté RL.
   2. However, real-world constraints are most non-discounted. For example,
      1. Constrain the cumulative cost of a company in the next year to be smaller than a specific value.
      2. Constrain the carbon emissions from the power grid in the next year to be smaller than a specific value.
   3. The discount factor is a way to approximate the original constraint to make RL method converge. Thus existing methods will transform the original constraint to discounted constraint in the code.
      1. This issue is also explained detailed in C2IQL [3].
   4. However, in this paper, the constraint is originally defined as discounted and all experiments are evaluated on discounted value.
      1. Why does this paper formulated in this way?
      2. Why not estimate on discounted value and test on non-discounted value since real-world constraints tend to be non-discounted like all previous works?
      3. Is there any gap or any influence?

[1] Sootla, A., Cowen-Rivers, A. I., Jafferjee, T., Wang, Z., Mguni, D. H., Wang, J., & Ammar, H. (2022, June). Sauté rl: Almost surely safe reinforcement learning using state augmentation. In *International Conference on Machine Learning* (pp. 20423-20443). PMLR.

[2] Castellano, A., Min, H., Mallada, E., & Bazerque, J. A. (2022, May). Reinforcement learning with almost sure constraints. In *Learning for Dynamics and Control Conference* (pp. 559-570). PMLR.

[3] Zifan, L. I. U., Li, X., & Zhang, J. C2IQL: Constraint-Conditioned Implicit Q-learning for Safe Offline Reinforcement Learning. In *Forty-second International Conference on Machine Learning*.

### Weakness on Experiments

1. Benchmarks are incomplete:
   1. For SafetyGymnasium, it is acceptable to select 4 tasks due to the large number of tasks.
   2. However, the following benchmarks are incomplete:
      1. Bullet-Safety-Gym contains 8 tasks while only 4 tasks are tested.
      2. Velocity dataset contains 5 tasks while only 2 are tested.
   3. The reason why all tasks in these benchmarks should be tested are:
      1. In one benchmark, different tasks reflects different feature or faces different difficulties. It is necessary to test them all to show the generality of proposed method.
      2. Thus, most existing papers (except for some early articles) test all tasks if corresponding benchmarks are selected.
      3. Test all tasks in one benchmark is more fair because it can eliminate the suspicion of selecting suitable environments.

2. Baselines are a little old and need to be updated.
   1. The baselines are mainly from 2019-2023, which are too old since there are lots of method proposed within 2 years.

   2. Here are some related methods
      1. Methods related to generative model and feasibility:
         1. FISOR [4], which focus on feasibility and hard constraint on zero-cost.
         2. OASIS [5], which focus on generating feasible trajectories.

      2. Methods related to broad range budgets:
         1. CAPS [6]
         2. CCAC [7]

   3. Why CDT is modified to $\gamma$-CDT?
      1. CDT is designed to non-discounted cost via bypassing the Bellman backup with discount factor. This makes CDT closer to real-world non-discounted constraint.
      2. How about inputting the discounted value as non-discounted value to CDT directly?

3. It is suggested to test the performance on non-discounted value as all previous works did.
   1. It is unrealistic to regard discounted values as constraints considering real-world constraints are most non-discounted.

   2. No matter in RL or Safe RL, the objective is always maximizing non-discounted cumulative rewards or constraining non-discounted cumulative costs.
      1. The discount factor is merely a compromise made by RL for convergence and stability.

      2. In RL, even if the value function estimates discounted cumulative rewards, the final performance is still evaluated on non-discounted cumulative rewards.

4. While this paper explained why new metrics are proposed, it is not clear on comparison and evaluation. Here are some suggestions.
   1. The primary evaluation criteria on normalized costs and rewards should also be included:
      1. It can more intuitively reflect the performance of the algorithm.
      2. It can be cross-validated with the results from other papers.
   2. More explanation on FR and CRR should be included:
      1. What is the meaning of "rate of generated trajectories"
         1. Methods like BCQ-Lag, CDT do not generate any trajectory
      2. Why you utilize FR and CRR? What is the advantage?
         1. For example:
            1. why this paper calculates discounted value for CRR?
               1. why not non-discounted value?
            2. why this paper only focus on the performance of feasible trajectories?

5. An ablation study on applying AEGIS to different policy structures is needed to demonstrate
   1. Why is diffusion policy the most suitable policy?
   2. The problem of other policy and the reason for the problem.
   3. Is AEGIS generalizable to different policy structures?

[4] Zheng, Y., Li, J., Yu, D., Yang, Y., Li, S. E., Zhan, X., & Liu, J. (2024). Safe offline reinforcement learning with feasibility-guided diffusion model. *arXiv preprint arXiv:2401.10700*.

[5] Yao, Y., Cen, Z., Ding, W., Lin, H., Liu, S., Zhang, T., ... & Zhao, D. (2024). Oasis: Conditional distribution shaping for offline safe reinforcement learning. *Advances in Neural Information Processing Systems*, *37*, 78451-78478.

[6] Chemingui, Y., Deshwal, A., Wei, H., Fern, A., & Doppa, J. (2025, April). Constraint-adaptive policy switching for offline safe reinforcement learning. In *Proceedings of the AAAI Conference on Artificial Intelligence* (Vol. 39, No. 15, pp. 15722-15730).

[7] Guo, Z., Zhou, W., Wang, S., & Li, W. (2025, March). Constraint-conditioned actor-critic for offline safe reinforcement learning. In *The Thirteenth International Conference on Learning Representations*.

**Questions:**

Please refer to Weaknesses above.

---

> ### Author Response · Authors · 2025-11-25
>
> Thank you for your review. We address your points below.
>
> ## Weakness on Problem Setup
>
> Like Sootla et al. [1] and Lin et al. [8], we formulate our problem based on discounted returns. Using discounted value is standard in (safe) reinforcement learning, and several works interpret this as modeling time preference [9] or an interest rate [10].
>
> In almost surely safe RL under this setting, the remaining budget is defined with respect to the discounted cost return, so evaluating feasibility with a non-discounted constraint would be inconsistent. Therefore, we also evaluate feasibility and performance based on the discounted cost return. In our analysis and implementation, we assume $0 <\gamma<1$, which is the standard discounted setting and ensures bounded returns.
>
> We agree that constraints on the non-discounted cost return are important in many real-world applications. We thank the reviewer for emphasizing this perspective. Extending AEGIS to problems with non-discounted constraints would be an interesting direction for future work. Possible approaches include: (i) following C2IQL [3] and learning a mapping from discounted returns to equivalent non-discounted values, or (ii) deriving heuristic mappings between non-discounted and discounted budgets under assumptions such as time-homogeneous costs. We will explicitly discuss this limitation and these possible extensions in the limitation section.
>
> ---
>
> [1] Sootla, A., Cowen-Rivers, A. I., Jafferjee, T., Wang, Z., Mguni, D. H., Wang, J., & Ammar, H. (2022, June). Sauté rl: Almost surely safe reinforcement learning using state augmentation. In International Conference on Machine Learning (pp. 20423-20443). PMLR.
>
> [3] Zifan, L., Li, X., & Zhang, J. C2IQL: Constraint-Conditioned Implicit Q-learning for Safe Offline Reinforcement Learning. In Forty-second International Conference on Machine Learning.
>
> [8] Lin, Q., Tang, B., Wu, Z., Yu, C., Mao, S., Xie, Q., Wang, X., & Wang, D. (2023, July). Safe offline reinforcement learning with real-time budget constraints. In Proceedings of the 40th International Conference on Machine Learning (PMLR, Vol. 202, pp. 21127–21152).
>
> [9] Schultheis, M., Rothkopf, C. A., & Koeppl, H. (2022). Reinforcement learning with non-exponential discounting. In Advances in Neural Information Processing Systems (NeurIPS 2022), 35.
>
> [10] Kaelbling, L. P., Littman, M. L., & Moore, A. W. (1996). Reinforcement learning: A survey. Journal of Artificial Intelligence Research, 4, 237–285.

---

> ### Author Response · Authors · 2025-11-25
>
> ## Weakness on Experiments
>
> ### 1. Completeness of benchmark
>
> As described in the general response, we have added the Walker2d Vel and CarCircle tasks. With these additions, we now cover all tasks listed in Table 3 ("Evaluation results of the normalized reward and cost") of the DSRL benchmark paper for SafetyGymnasium-Velocity and BulletSafetyGym. Thus, our task selection follows the benchmark paper rather than being chosen by us ad hoc.
>
> The DSRL paper also introduces additional tasks whose empirical results are reported only in its appendix. Due to computational constraints during the rebuttal period, we were not able to run the remaining tasks (Hopper Vel, Ant Vel, BallRun, BallCircle, and CarRun). We will include them in the camera-ready version if the paper is accepted.
>
> ### 2. Baselines
>
> We appreciate the reviewer for pointing out additional baselines. Comparing with the latest methods would certainly be valuable, but there are several obstacles to a fair comparison within our current setting.
> FISOR focuses on a cost limit of zero without using a budget parameter, whereas our setting explicitly conditions on a budget. Adapting FISOR to this budget setting is non-trivial.
> OASIS uses a generative model to augment the dataset. We view this as largely orthogonal and potentially complementary to AEGIS, since similar augmentation could also be applied on top of our method.
> CAPS and CCAC are based on non-discounted, expectation-type constraints, whereas AEGIS focuses on discounted almost-sure constraints. Making these methods directly comparable in our discounted almost-sure setting would again require redesign of the evaluation protocol.
>
> We agree that including these baselines (or suitably adapted versions) would strengthen the empirical study, and we will consider them as future work beyond the scope of the rebuttal-period experiments.
>
> > How about inputting the discounted value as non-discounted value to CDT directly?
>
> If we understand your question correctly, this is what our $\gamma$-CDT baseline does. In $\gamma$-CDT, we compute the discounted cost return from each offline trajectory and feed it to CDT as the cost return-to-go, so that CDT uses the same discounted constraint definition as AEGIS.
>
> ---
>
> [4] Zheng, Y., Li, J., Yu, D., Yang, Y., Li, S. E., Zhan, X., & Liu, J. (2024). Safe offline reinforcement learning with feasibility-guided diffusion model. arXiv preprint arXiv:2401.10700.
>
> [5] Yao, Y., Cen, Z., Ding, W., Lin, H., Liu, S., Zhang, T., ... & Zhao, D. (2024). Oasis: Conditional distribution shaping for offline safe reinforcement learning. Advances in Neural Information Processing Systems, 37, 78451-78478.
>
> [6] Chemingui, Y., Deshwal, A., Wei, H., Fern, A., & Doppa, J. (2025, April). Constraint-adaptive policy switching for offline safe reinforcement learning. In Proceedings of the AAAI Conference on Artificial Intelligence (Vol. 39, No. 15, pp. 15722-15730).
>
> [7] Guo, Z., Zhou, W., Wang, S., & Li, W. (2025, March). Constraint-conditioned actor-critic for offline safe reinforcement learning. In The Thirteenth International Conference on Learning Representations.

---

> ### Author Response · Authors · 2025-11-25
>
> ### 3. Performance on non-discounted value
>
> We agree that constraints on non-discounted cumulative costs are often important in practice. At the same time, we respectfully do not fully share the view that the discount factor is *merely* a compromise for convergence, as discussed above and in prior work [9,10]. Moreover, some empirical studies in RL and safe RL evaluate algorithms using discounted returns as their primary performance metric [11,12].
>
> ### 4. New metrics
>
> > The primary evaluation criteria on normalized costs and rewards should also be included.
>
> Thank you for the suggestion. In the revised version, we will add (discounted) normalized cost and reward returns as additional evaluation metrics in the appendix.
>
> > What is the meaning of "rate of generated trajectories"?
>
> By “generated trajectories,” we mean the trajectories obtained by rolling out the learned agent in the evaluation environment. We apologize for the ambiguity and will make this clearer in the text.
>
> > Why does this paper calculate discounted value for CRR? Why not non-discounted value?
>
> As discussed above, feasibility in our setting is defined using the discounted cost return. If we were to evaluate the reward using a non-discounted return while still constraining discounted cost, the time preference for reward and cost would become asymmetric. For example, an agent that behaves conservatively early and then aggressively late in the episode would appear overly favorable under such an evaluation. Using discounted returns for both reward and cost avoids this asymmetry.
>
> > Why does this paper only focus on the performance of feasible trajectories?
>
> Our intention with CRR is to measure performance subject to satisfying the constraint. Standard (unconstrained) reward returns would favor infeasible-but-high-reward trajectories. However, such trajectories have failed the safety requirements, so counting their rewards as a performance metric would be misleading. That said, to reflect the reviewer’s concern, we will also include the (unconstrained) discounted reward return in the appendix.
>
> ### 5. Different policy structures
>
> We chose diffusion because it is a strong, widely used policy class in offline RL. We also view AEGIS's applicability to diffusion policies as one of its practical advantages. Extending the same critic-guided weighting to other policy classes would be possible and interesting, but we leave a full empirical study for future work.
>
> ---
>
> [11] Patterson, A., Neumann, S., White, M., & White, A. (2024). Empirical design in reinforcement learning. Journal of Machine Learning Research, 25(318), 1–63.
>
> [12] Hazra, S., Dasgupta, P., & Dey, S. (2025). Incentivizing safer actions in policy optimization for constrained reinforcement learning. In Proceedings of the Thirty-Fourth International Joint Conference on Artificial Intelligence (IJCAI-25) (pp. 5318–5326).

---

> > ### Comment · Reviewer_rtx1 · 2025-11-26
> >
> > Thank you very much for the authors’ brief responses. These replies have addressed few of my concerns, but not enough. Here are some comments for the rebuttal content.
> >
> > ### **Q1: About Benchmark Selection**
> >
> > 1. BulletGym only has 8 tasks; this paper selects parts of them while most existing papers in OSRL test all of them. Even if I mentioned this point, only CarCircle is included. **There is a serious suspicion of cherry picking rather than a general evaluation.**
> > 2. Running DSRL tasks does not need much computational resources. Besides, there are a lot of days left for rebuttal. What is the meaning of " Due to computational constraints during the rebuttal period, we were not able to run the remaining tasks"?
> >
> > ### **Q2: About baselines**
> >
> > 1. This paper includes BCQ-L, COptiDICE, CDT, and TREBI for comparison. This indicates that AEGIS can compare with common OSRL algorithms. However, CAPS and CCAC are all OSRL algorithms that focus on the same problem of BCQ-L, COptiDICE, and CDT. Thus, it is unreasonable that these methods cannot be compared.
> > 2. I understand that FISOR focuses on a cost limit of zero. However, recent work on OSRL usually includes it in comparison to validate "How the proposed algorithm performs better than this extreme situation (FISOR)".
> > 3. All baselines, which I think should be included, provide the code in the paper or in supplementary materials. However, the authors are unwilling to run these codes.
> >
> > ### **Q3: About discounted formulation and new metrics.**
> >
> > 1. I understand that this paper is built based on the discounted formulation. However, I am just curious about the performance on **non-discounted cumulative reward/cost returns**. I want to see these results because **all existing OSRL papers are evaluated based on these two metrics**.
> > 2. I also understand that this paper utilized new metrics in this paper. However, it is unreasonable to only show these new metrics, which have never (at least seldom) been shown in existing OSRL papers. Compared to these two metrics proposed in this paper, the **non-discounted cumulative reward/cost returns** are widely recognized metrics in all OSRL papers.
> >
> > Taking all the responses into account, I am inclined to keep my score.

---

> > > ### Author Response · Authors · 2025-12-03
> > >
> > > We thank the reviewer for their response and address the additional questions below.
> > >
> > > To address the reviewer’s concern about benchmark task selection, we add the results of $\gamma$-CDT, TREBI, IDQL-Saut\'e, and ours (AEGIS) on the Ant Vel, Hopper Vel, CarRun, BallRun, and BallCircle tasks, and we report the updated averages (including other tasks reported in the general response) below. With these additions, we now cover all eight BulletGym tasks and five SafetyGymnasium-Velocity tasks.
> > >
> > > |TASK|$\gamma$-CDT|TREBI|IDQL-Saut\'e|Ours|
> > > |:---|:---:|:---:|:---:|:---:|
> > > ||FR$\uparrow$ \\ CRR$\uparrow$|FR$\uparrow$ \\ CRR$\uparrow$|FR$\uparrow$ \\ CRR$\uparrow$|FR$\uparrow$ \\ CRR$\uparrow$|
> > > |Ant Vel|0.03 \\ 0.28|0.69 \\ 0.08|0.99 \\ 0.86|1.00 \\ 0.74|
> > > |Hopper Vel|0.37 \\ 0.05|0.25 \\ 0.12|0.78 \\ 0.22|0.80 \\ 0.27|
> > > |**SafetyGymAvg.**|**0.60** \\ **0.40**|**0.52** \\ **0.29**|**0.67** \\ **0.37**|**0.89** \\ **0.37**|
> > >
> > > |TASK|$\gamma$-CDT|TREBI|IDQL-Saut\'e|Ours|
> > > |:---|:---:|:---:|:---:|:---:|
> > > ||FR$\uparrow$ \\ CRR$\uparrow$|FR$\uparrow$ \\ CRR$\uparrow$|FR$\uparrow$ \\ CRR$\uparrow$|FR$\uparrow$ \\ CRR$\uparrow$|
> > > |CarRun|0.93 \\ 0.99|0.79 \\ 0.99|0.68 \\ 0.39|1.00 \\ 0.93|
> > > |BallRun|1.00 \\ 0.52|0.22 \\ 0.38|0.99 \\ -0.01|0.34 \\ -0.12|
> > > |BallCircle|0.44 \\ 0.71|0.78 \\ 0.77|0.81 \\ 0.10|1.00 \\ 0.50|
> > > |**BulletGymAvg.**|**0.56** \\ **0.64**|**0.49** \\ **0.44**|**0.79** \\ **0.20**|**0.86** \\ **0.40**|
> > >
> > > ---
> > >
> > > In the discounted formulation, if the agent incurs a small cost early in the episode, the remaining budget increases accordingly. As a result, agents can accumulate a relatively large non-discounted cumulative cost while still satisfying the (discounted) constraint. AEGIS, which conditions its decisions on the remaining budget, can therefore also exhibit high non-discounted cumulative cost, but this does not reflect infeasibility in our formulation.

---

### Official Review · Reviewer_rStD · 2025-11-01

**Soundness:** 2
**Presentation:** 3
**Contribution:** 2
**Rating:** 4
**Confidence:** 4

**Summary:**

This paper studies almost sure safety in offline RL. The authors consider the cost budget-augmented MDP, which augments the state $s$ to $\tilde{s}\doteq(s,\delta)$, and derive the feasibility V and Q value function along with the Bellman operator. In short, the feasibility value function is used to measure whether an augmented state (or state-action pair) is feasible to satisfy the constraint. The authors propose to use expectile regression to estimate the feasibility value and further employ a diffusion model as final policy. The experiment on offline safe RL benchmark shows that the proposed AEGIS method exceeds baselines in terms of almost sure safety.

**Strengths:**

- The paper is clearly written. The preliminaries and related background are sufficiently introduced.
- Overall the method is novel. The use of feasibility function converts the trajectory-wise safety constraint satisfaction to transition-level value estimation. The use of expectile regression makes it a good estimation according to the infimum in definition.
- The experiment shows that AEGIS has better almost sure safety.

**Weaknesses:**

- While most contents of methodology (sec.4.1~4.3) focus on feasibility and reward value function, the final policy uses a diffusion model. I believe the key contribution of this paper is on the value function part instead of a diffusion policy. Therefore, it is necessary to run an ablation of removing diffusion policy. Additionally, it will make the performance comparison in table 1 more fair as most baselines are not using diffusion policy.
- In table 1, though AEGIS achieves higher feasibility rate, the constrained reward return is much lower than many baselines. In other words, the policy of AEGIS is safer but also more conservative. In this case, it's hard to say AEGIS is better especially many baselines are not designed for almost sure safety setting.

Minor issues:
- line 69, 18%p -> 18%

**Questions:**

- Do you use the same or different hyperparameters (e.g., reward penalty n, $\tau$, and others in diffusion model) for each task?
- The reward value function takes $\delta$ as input. However, when training the reward value functions, AEGIS uniformly samples one $\delta$. Why do you use uniform sampling? Does this mean for each state-action pair $(s,a)$ the Q or V value will only be trained on one $\delta$?

---

> ### Author Response · Authors · 2025-11-23
>
> Thank you for your review. We address your points below.
>
> ## Weakness 1: Diffusion policy
> Several baselines already use diffusion models or expressive sequence models: TREBI and IDQL-Saut\'e use diffusion models and $\gamma$-CDT uses a transformer architecture. Among these, IDQL-Saut\'e is particularly close to AEGIS in terms of diffusion policy, including network structure, so our comparison with IDQL-Saut\'e highlights the effect of our feasibility/reward critic design rather than diffusion itself.
>
> It is true that our theoretical analysis and algorithms (except for policy training) are not specific to diffusion. We chose diffusion because it is a strong, widely used policy class in offline RL. We also view AEGIS's applicability to diffusion policies as one of its practical advantages. Extending the same critic-guided weighting to other policy classes would be possible and interesting, but we leave a full empirical study for future work.
>
> ## Weakness 2: FR-CRR trade-off
> Safety and performance are often in a trade-off: an agent that acts more conservatively to increase FR will naturally see some decrease in CRR. In our setting, however, an agent that frequently violates the constraint while achieving high reward is less favorable from the problem formulation. Note also that method targeting almost-sure or trajectory-level constraints (γ-CDT, TREBI, IDQL-Sauté) also struggle to obtain high FR on many tasks, indicating that achieving high feasibility is genuinely challenging.
>
> As reported in the general response, we provide additional results for the comparison with expectation-constrained baselines (BCQ-Lag and COptiDICE). The results show that, AEGIS agent can achieve reward performance comparable to or better than these methods, while still achieving better worst-case safety (lower P90 cost). For further details, please refer to the general response.
>
> ## Weakness 3: Minor issues
> The reported "18%p" refers to the difference between feasibility rates 69% (0.69) and 88% (0.88). In the updated table, it is 22 percentage points between 68% and 90%. To avoid confusion, we will replace "%p" with "percentage points" in the revised version.
>
> ## Question 1: Hyperparameters
> As mentioned around line 395 and in the limitation section, we use a single set of AEGIS hyperparameters uniformly across all tasks in Table 1. The values (including the $n$ and $\tau$) are summarized in Appendix D.2, Table 2. We also provide a sensitivity ablation on $\alpha$ and $n$ in Appendix E.1., evaluated per task.
>
> ## Question 2: Sampling $\delta$
> The reward critics take the current remaining budget $\delta$ (not the initial budget $\delta_0$) as input. During training, to make efficient use of a finite offline dataset and to cover a wide range of budgets, we relabel sampled transitions with random $\delta$. Over the course of training, each $(s,a)$ pair is thus seen under various $\delta$ values, rather than only at a single $\delta$ for each $(s,a)$.

---

### Official Review · Reviewer_omGa · 2025-11-01

**Soundness:** 3
**Presentation:** 2
**Contribution:** 2
**Rating:** 4
**Confidence:** 2

**Summary:**

The paper proposes AEGIS, an offline safe RL method aiming at almost-sure safety across safety budgets. The key idea is to view the feasible state-budget pairs as augmented states to show that there exists an optimal policy for the feasible set, and feasible augmented states are the epigraph of a feasibility critic that can be learned offline via an expectile-based Bellman update. Then a diffusion policy is guided by both this feasibility critic an a reward critic, so that sampled actions are within budget and high-value. Empirical studies on DSRL and a humanoid task report higher feasibility rates than several baselines.

**Strengths:**

1. The paper addresses almost-sure constraint, a harder notion of safety, in offline RL, which is under-explored.
2. The epigraph construction of state-budget pairs offers a clean perspective of tackling the problem with Bellman operators.

**Weaknesses:**

1. The authors repeatedly claims "almost-surely safe offline RL", yet the method approximates the worst-case with a finite-$\alpha$ expectile and a finite penalty. The experiments show many tasks with feasibility rates well below 1.0 (c.f. table 1).
2. The feasibility operator is defined with an essential sup, but the practical algorithm instead adopts an expectile and then trains a diffusion policy with guidance. The text notes that the two methods collide as $\alpha\rightarrow 1$. However, there is no finite-sample guarantee that the implemented $\alpha$ and guidance deliver almost-sure feasibility.
3. The empirical results are thin and not compelling. Gains in feasibility rates often come with reduced constrained return.
4. The method relies on learning a worst-case feasibility critic from a fixed dataset, but offline data can rarely cover the unsafe tails needed for an ess sup backup. I am curious how would the author improve the robustness against narrow missspecified or narrow datasets?

**Questions:**

Please see weaknesses.

---

> ### Author Response · Authors · 2025-11-23
>
> Thank you for your review. We address your points below.
>
> ## Weakness 1 and 2: almost-sure guarantees
> We fully agree that our implemented algorithm does not provide a finite-sample almost-sure guarantee. Our work theoretically analyzes the feasibility of almost surely safe RL, and this is what motivates the term "almost surely safe offline RL." As we already state in the paper, the practical algorithm relies on expectile approximation (with $\alpha<1$ and $n<\infty$), and therefore does not provide an almost-sure guarantee. However, by focusing on the worst case, our empirical results show that it achieves a higher feasibility rate than the baselines.
>
> We appreciate the reviewer for highlighting the finite-sample aspect. We will expand the limitation section to explicitly note the finite offline dataset and limited expressiveness of the neural network as sources that prevent guarantees, in addition to the expectile approximation already discussed. We will also carefully revisit the abstract and introduction to soften any wording that could be interpreted as claiming a practical almost-sure guarantee.
>
> ## Weakness 3: empirical results
> The DSRL benchmark is a recently proposed benchmark for offline safe RL, which offers a diverse range of tasks. We therefore adopt it as our primary benchmark. In the general response, we additionally report results on the Walker2dVel and CarCircle tasks, further supporting the empirical trends reported in the main paper.
>
> Safety and performance are often in a trade-off: an agent that acts more conservatively to increase FR will naturally see some decrease in CRR. In our setting, however, an agent that frequently violates the constraint while achieving high reward is less favorable from the problem formulation. Note also that methods targeting almost-sure or trajectory-level constraints (γ-CDT, TREBI, IDQL-Sauté) also struggle to obtain high FR on many tasks, indicating that achieving high feasibility is genuinely challenging.
>
> As reported in the general response, we provide additional results for the comparison with expectation-constrained baselines (BCQ-Lag and COptiDICE). The results show that, AEGIS agent can achieve reward performance comparable to or better than these methods, while still achieving better worst-case safety (lower P90 cost). For further details, please refer to the general response.
>
> ## Weakness 4: coverage of unsafe tails
> We fully agree that our method cannot capture dangers that are never observed in the offline dataset, which is inherent to offline methods without additional supervision. Our feasibility critic is instead designed to emphasize worst-case (tail) outcomes within the observed transition distribution. In practical applications, one possible way to further improve robustness is to collect additional trajectories that target anticipated risk modes, which we leave for future work.

---

### Author Response · Authors · 2025-11-23
**General Response: Additional Experimental Results**

We sincerely thank all reviewers for their constructive feedback. In this general response, we report additional experimental results that address issues raised in multiple reviews.

---

Below, we provide an updated version of Table 1, where we add the Walker2d Vel and CarCircle tasks. With these additions, AEGIS achieves FR$\ge 0.75$ on 11 of 12 tasks, while maintaining competitive CRR.

### SafetyGym tasks

|TASK|BCQ-Lag|COptiDICE|$\gamma$-CDT|TREBI|IDQL-Saut\'e|Ours|
|:---|:---:|:---:|:---:|:---:|:---:|:---:|
||FR$\uparrow$ \\ CRR$\uparrow$|FR$\uparrow$ \\ CRR$\uparrow$|FR$\uparrow$ \\ CRR$\uparrow$|FR$\uparrow$ \\ CRR$\uparrow$|FR$\uparrow$ \\ CRR$\uparrow$|FR$\uparrow$ \\ CRR$\uparrow$|
|CarButton1|0.41 \\ 0.24|0.58 \\ 0.15|0.42 \\ 0.24|0.47 \\ 0.26|0.45 \\ 0.15|0.79 \\ 0.10|
|CarGoal1|0.69 \\ 0.48|0.65 \\ 0.49|0.87 \\ 0.39|0.57 \\ 0.52|0.45 \\ 0.46|0.95 \\ 0.18|
|PointButton1|0.43 \\ 0.30|0.51 \\ 0.23|0.53 \\ 0.24|0.53 \\ 0.25|0.50 \\ 0.17|0.80 \\ 0.12|
|PointGoal1|0.56 \\ 0.59|0.58 \\ 0.41|0.81 \\ 0.36|0.67 \\ 0.53|0.40 \\ 0.50|0.75 \\ 0.18|
|HalfCheetah Vel|0.07 \\ 0.87|1.00 \\ 0.59|0.21 \\ 0.40|1.00 \\ 0.35|1.00 \\ 0.33|1.00 \\ 0.84|
|Swimmer Vel|0.01 \\ 0.09|0.15 \\ 0.27|0.99 \\ 0.41|0.00 \\ -|0.95 \\ 0.05|0.91 \\ 0.20|
|Walker2d Vel|0.99 \\ 0.77|0.33 \\ 0.50|0.35 \\ 0.73|0.39 \\ 0.14|0.48 \\ 0.58|1.00 \\ 0.69|
|**SafetyGymAvg.**|**0.45** \\ **0.48**|**0.54** \\ **0.38**|**0.60** \\ **0.40**|**0.52** \\ **0.29**|**0.60** \\ **0.32**|**0.89** \\ **0.33**|

### BulletGym tasks

|TASK|BCQ-Lag|COptiDICE|$\gamma$-CDT|TREBI|IDQL-Saut\'e|Ours|
|:---|:---:|:---:|:---:|:---:|:---:|:---:|
||FR$\uparrow$ \\ CRR$\uparrow$|FR$\uparrow$ \\ CRR$\uparrow$|FR$\uparrow$ \\ CRR$\uparrow$|FR$\uparrow$ \\ CRR$\uparrow$|FR$\uparrow$ \\ CRR$\uparrow$|FR$\uparrow$ \\ CRR$\uparrow$|
|AntRun|0.07 \\ 0.36|0.92 \\ 0.61|0.16 \\ 0.72|0.30 \\ 0.70|0.99 \\ 0.68|1.00 \\ 0.71|
|AntCircle|0.35 \\ 0.49|0.50 \\ 0.14|0.04 \\ 0.17|0.57 \\ 0.04|0.98 \\ 0.04|1.00 \\ 0.03|
|DroneRun|0.08 \\ 0.66|0.00 \\ -|0.91 \\ 0.75|0.30 \\ 0.04|0.69 \\ 0.08|0.55 \\ 0.32|
|DroneCircle|0.10 \\ 0.68|0.78 \\ 0.40|0.22 \\ 0.68|0.03 \\ 0.03|0.50 \\ 0.27|0.99 \\ 0.51|
|CarCircle|0.67 \\ 0.40|0.46 \\ 0.34|0.79 \\ 0.58|0.89 \\ 0.59|0.71 \\ 0.02|1.00 \\ 0.35|
|**BulletGymAvg.**|**0.25** \\ **0.52**|**0.53** \\ **0.30**|**0.42** \\ **0.58**|**0.42** \\ **0.28**|**0.77** \\ **0.22**|**0.91** \\ **0.38**|

---

Since expectation constraints are less conservative than almost-sure constraints for the same constraint threshold, we have conducted additional experiments to compare AEGIS with those baselines (BCQ-Lag and COptiDICE).

1. We train BCQ-Lag and COptiDICE with an expectation constraint threshold set to half of the target almost-sure constraint threshold and evaluate the resulting agents under the original (unhalved) target threshold. Even under this more conservative configuration, their feasibility rates remain low on several tasks.

### Conservative expectation constraint threshold
|TASK|BCQ-Lag (conservative)|COptiDICE (conservative)|
|:---|:---:|:---:|
||FR$\uparrow$ \\ CRR$\uparrow$|FR$\uparrow$ \\ CRR$\uparrow$|
|AntRun|0.05 \\ 0.38|0.96 \\ 0.61|
|AntCircle|0.41 \\ 0.49|0.37 \\ 0.13|
|DroneRun|0.03 \\ 0.69|0.00 \\ -|
|DroneCircle|0.39 \\ 0.66|0.75 \\ 0.37|
|CarCircle|0.75 \\ 0.39|0.36 \\ 0.34|
|**BulletGymAvg.**|**0.33** \\ **0.52**|**0.49** \\ **0.29**|

2. For AEGIS, we also evaluate the learned agents under higher cost thresholds than in the main experiments and report the 0.9 quantile (P90) of the discounted cost return and the mean discounted reward return. These results show that, for more relaxed thresholds at inference time, AEGIS can achieve performance similar to or higher than BCQ-Lag and COptiDICE, while still maintaining better worst-case safety (lower P90 cost).

### BCQ-Lag vs Ours (high $\delta_0$)

|TASK|BCQ-Lag|Ours (high $\delta_0$)|
|:---|:---:|:---:|
||P90 cost return$\downarrow$ \\ reward return$\uparrow$|P90 cost return$\downarrow$ \\ reward return$\uparrow$|
|AntRun|41.97 \\ 0.72|31.91 \\ 0.80|
|AntCircle|17.85 \\ 0.54|16.00 \\ 0.42|
|DroneRun|35.71 \\ 0.68|15.40 \\ 0.79|
|DroneCircle|22.59 \\ 0.79|19.85 \\ 0.71|
|CarCircle|12.27 \\ 0.46|8.86 \\ 0.48|
|**BulletGymAvg.**|**26.08** \\ **0.64**|**18.40** \\ **0.64**|

### COptiDICE vs Ours (high $\delta_0$)

|TASK|COptiDICE|Ours (high $\delta_0$)|
|:---|:---:|:---:|
||P90 cost return$\downarrow$ \\ reward return$\uparrow$|P90 cost return$\downarrow$ \\ reward return$\uparrow$|
|AntRun|7.79 \\ 0.61|3.55 \\ 0.71|
|AntCircle|25.32 \\ 0.21|12.39 \\ 0.30|
|DroneRun|15.94 \\ 0.53|15.40 \\ 0.79|
|DroneCircle|9.03 \\ 0.41|5.63 \\ 0.52|
|CarCircle|30.18 \\ 0.38|8.86 \\ 0.48|
|**BulletGymAvg.**|**17.65** \\ **0.43**|**9.17** \\ **0.56**|

---

### Meta-Review · Area_Chair_MTR5 · 2025-12-03

**Summary:**

This paper proposes a safe offline RL algorithm called AEGIS. It characterizes the feasible set of initial state-budget pairs as the epigraph of a feasibility critic updated via the worst-case backup, and then extends IQL-style expectile regression loss to train feasibility and reward critics. Lastly, the policy is extracted via weighted diffusion loss. The overall framework shares lots of similarity with FISOR (Zheng et al. 2024, cited in the paper, but is not sufficiently discussed or compared), except for the addition of extra budget and the use of discounted cost.

The reviewers have raised many concerns, including:
- Some level of overclaiming: claiming "almost-surely safe", yet approximates the worst-case with a finite-expectile and a finite penalty (Reviewer omGa)
- Lacks sufficient acknowledgement and discussion on highly relevant works (Reviewer rtx1, CpiR).
- Necessity of the discounted cost setup (Reviewer rtx1)
- Need an extra reward penalty, which could be impractical to set (Reviewer CpiR)
- Weak experiment: missing lots of highly relevant baselines (Reviewer rtx1, CpiR); over-conservative performance (Reviewer omGa, rStD); lack of sufficient ablation (Reviewer rStD, rtx1).

I've read the author's rebuttal, but feel that many of the reviewers' concerns are still not well addressed. Therefore, I think the paper in its current shape is not ready to be accepted.

**Reviewer Concerns:**

The reviewers' concerns that remain unresolved:
- Some level of overclaiming: claiming "almost-surely safe", yet approximates the worst-case with a finite-expectile and a finite penalty (Reviewer omGa)
- Lacks sufficient acknowledgement and discussion on highly relevant works (Reviewer rtx1, CpiR).
- Necessity of the discounted cost setup (Reviewer rtx1)
- Need an extra reward penalty, which could be impractical to set (Reviewer CpiR)
- Weak experiment: missing lots of highly relevant baselines (Reviewer rtx1, CpiR);  (Reviewer rtx1); over-conservative performance (Reviewer omGa, rStD); lack of sufficient ablation (Reviewer rStD, rtx1).

Reviewer rtx1 also has concerns regarding the paper reporting results only on part of the tasks in DSRL, which raises suspicions of cherry picking. I think this concern has been partly resolved based on the new results provided by the authors during rebuttal.

**Reviewer Scores:**

I don't think reviewer omGa, rtx1, and CpiR will change their scores. Reviewer CpiR clearly responded that he will maintain his original score.

Reviewer rStD might change the score, but I think he/she will more likely maintain the original evaluation.

---

### Decision · Program_Chairs · 2026-01-26

Reject